



# Ozone dry deposition through plant stomata: Multi-model comparison with flux observations and the role of water stress as part of AQMEII4 Activity 2

Anam M. Khan[1], Olivia E. Clifton[2,3], Jesse O. Bash[4], Sam Bland[5], Nathan Booth[6], Philip Cheung[7], Lisa Emberson[6], Johannes Flemming[8], Erick Fredj[9], Stefano Galmarini[10], Laurens Ganzeveld[11], Orestis Gazetas[10,a], Ignacio Goded[10], Christian Hogrefe[4], Christopher D. Holmes[12], László Horváth[13], Vincent Huijnen[14], Qian Li[15], Paul A. Makar[7], Ivan Mammarella[16], Giovanni Manca[10], J. William Munger[17,18], Juan L. Pérez-Camanyo[19], Jonathan Pleim[20], Limei Ran[21], Roberto San Jose[19], Donna Schwede[4], Sam J. Silva[22], Ralf Staebler[7], Shihan Sun[23], Amos P.K. Tai[23,24,25], Eran Tas[15], Timo Vesala[16,26], Tamás Weidinger[27], Zhiyong Wu[28,b], Leiming Zhang[7], and Paul C. Stoy[29]

[1]Department of Forest and Wildlife Ecology, University of Wisconsin-Madison, Madison, WI, USA
[2]NASA Goddard Institute for Space Studies, New York, NY, USA
[3]Center for Climate Systems Research, Columbia Climate School, Columbia University in the City of New York, New York, NY, USA
[4]Office of Research and Development, United States Environmental Protection Agency, Research Triangle Park, NC, USA
[5]Stockholm Environment Institute, Environment and Geography Department, University of York, York, UK
[6]Environment and Geography Department, University of York, York, UK
[7]Air Quality Research Division, Atmospheric Science and Technology Directorate, Environment and Climate Change Canada, Toronto, Canada
[8]European Centre for Medium-Range Weather Forecasts, Reading, UK
[9]Department of Computer Science, The Jerusalem College of Technology, Jerusalem, Israel
[10]Joint Research Centre (JRC), European Commission, Ispra, Italy
[11]Meteorology and Air Quality, Wageningen University, Wageningen, the Netherlands
[12]Department of Earth, Ocean and Atmospheric Science, Florida State University, Tallahassee, FL, USA
[13]ELKH-SZTE Photoacoustic Research Group, Department of Optics and Quantum Electronics, University of Szeged, Szeged, Hungary
[14]Royal Netherlands Meteorological Institute, De Bilt, the Netherlands
[15]The Institute of Environmental Sciences, The Robert H. Smith Faculty of Agriculture, Food and Environment, The Hebrew University of Jerusalem, Rehovot, Israel
[16]Institute for Atmospheric and Earth System Research/Physics, Faculty of Science, University of Helsinki, Helsinki, Finland
[17]School of Engineering and Applied Sciences, Harvard University, Cambridge, MA, USA
[18]Department of Earth and Planetary Sciences, Harvard University, Cambridge, MA, USA
[19]Computer Science School, Technical University of Madrid (UPM), Madrid, Spain
[20]Center for Environmental Measurement and Modeling, United States Environmental Protection Agency, Research Triangle Park, NC, USA
[21]Natural Resources Conservation Service, United States Department of Agriculture, Greensboro, NC, USA
[22]Department of Earth Sciences, University of Southern California, Los Angeles, CA, USA
[23]Department of Earth and Environmental Sciences, Faculty of Science, The Chinese University of Hong Kong, Hong Kong, China
[24]State Key Laboratory of Agrobiotechnology, The Chinese University of Hong Kong, Hong Kong, China
[25]Institute of Environment, Energy and Sustainability, The Chinese University of Hong Kong, Hong Kong, China



[26]Institute for Atmospheric and Earth System Research/Forest Sciences, Faculty of Agriculture and Forestry, University of Helsinki, Helsinki, Finland

[27]Department of Meteorology, Institute of Geography and Earth Sciences, Eötvös Loránd University, Budapest, Hungary

[28]ORISE Fellow at Center for Environmental Measurement and Modeling, United States Environmental Protection Agency, Research Triangle Park, NC, USA

[29]Biological Systems Engineering, University of Wisconsin-Madison, Madison, WI, USA

[a]now at: Scottish Universities Environmental Research Centre (SUERC), East Kilbride, UK

[b]now at: RTI International, Research Triangle Park, NC, USA

**Correspondence:** Anam M. Khan (amkhan7@wisc.edu)

**Abstract.** A substantial portion of tropospheric $O_3$ dry deposition occurs after diffusion of $O_3$ through plant stomata. Simulating stomatal uptake of $O_3$ in 3D atmospheric chemistry models is important in the face of increasing drought induced declines in stomatal conductance and enhanced ambient $O_3$. Here, we present a comparison of the stomatal component of $O_3$ dry deposition ($eg_s$) from chemical transport models and estimates of $eg_s$ from observed $CO_2$, latent heat, and $O_3$ flux. The dry

deposition schemes were configured as single-point models forced with data collected at flux towers. We conducted sensitivity analyses to study the impact of model parameters that control stomatal moisture stress on modeled $eg_s$. Examining six sites around the northern hemisphere, we find that the seasonality of observed flux-based $eg_s$ agrees with the seasonality of simulated $eg_s$ at times during the growing season with disagreements occurring during the later part of the growing season at some sites. We find that modeled water stress effects are too strong in a temperate-boreal transition forest. Some single-point models overestimate summertime $eg_s$ in a seasonally water-limited Mediterranean shrubland. At all sites examined, modeled $eg_s$

was sensitive to parameters that control the vapor pressure deficit stress. At specific sites that experienced substantial declines in soil moisture, the simulation of $eg_s$ was highly sensitive to parameters that control the soil moisture stress. The findings demonstrate the challenges in accurately representing the effects of moisture stress on the stomatal sink of $O_3$ during observed increases in dryness due to ecosystem specific plant-resource interactions.

## 1 Introduction

Tropospheric ozone ($O_3$) is a secondary air pollutant formed through photochemical reactions involving biogenic and anthropogenic emissions of methane, non-methane volatile organic compounds (VOC), or carbon monoxide (CO), and nitrogen oxides (NOx). Dry deposition of $O_3$ represents a substantial sink of tropospheric $O_3$ (Lelieveld and Dentener, 2000; Stevenson et al., 2006; Wild, 2007; Young et al., 2013), and the simulation of $O_3$ deposition velocity, $V_d$, can impact the simulation of $O_3$

concentrations near the surface propagating up the atmospheric vertical $O_3$ profile (Baublitz et al., 2020; Clifton et al., 2020b). Since dry deposition directly impacts ambient $O_3$ concentrations, modeling the multiple processes involved in dry deposition is an important component of 3D atmospheric chemistry models (Hardacre et al., 2015; Clifton et al., 2023). An important contribution to dry deposition occurs with the diffusion of $O_3$ through plant stomata, tiny pores on leaf surfaces which facilitate the exchange of gasses such as $CO_2$, $H_2O$, and $O_3$. $O_3$ can degrade to reactive oxygen species in the leaf apoplast and

react with compounds after stomatal uptake transports $O_3$ to the intercellular spaces of leaves (Baier et al., 2005; Dizengremel



et al., 2009; Wedow et al., 2021). The stomatal sink of $O_3$ makes stomatal conductance, determined largely by the aperture of stomatal pores and stomatal density, a key component of modeling dry deposition of $O_3$.

The ability of plants to sense environmental changes can result in osmotic adjustments that lead to changes in guard cell turgor, stomatal aperture, and eventually stomatal conductance, and through this mechanism, stomatal conductance can respond to changing environmental conditions on timescales as short as minutes (Hetherington and Woodward, 2003; Lawson and Vialet-Chabrand, 2019). Stomatal conductance responds to changes in soil moisture, photosynthetically active radiation (PAR), the vapor pressure deficit ($VPD$) between the interior of the leaf and the atmosphere, and intercellular $CO_2$ concentrations (Lawson, 2009; Lawson and Vialet-Chabrand, 2019; Grossiord et al., 2020; Matthews et al., 2020). $O_3$ uptake itself has been shown to impact stomatal conductance possibly through bursts of reactive oxygen species in guard cells and impacting biochemical pathways, including those involved in stomatal sensitivity to soil drying (Wilkinson and Davies, 2009, 2010; Vahisalu et al., 2010; Lombardozzi et al., 2012b). Many factors of global change such as rising ambient $CO_2$ concentrations, increasing temperatures, and the increasing severity and frequency of droughts (Dai, 2013; Zhao and Dai, 2022) are likely to impact stomatal conductance (Liang et al., 2023). Thus, in addition to the photosynthetic response, the stomatal response to global change could be an important pathway by which global change impacts major components of the carbon and the water cycle such as photosynthesis and transpiration (Hetherington and Woodward, 2003; Marchin et al., 2023). Therefore, simulating the stomatal response to the environment is a critical component of many land surface models and a large variety of stomatal conductance models are implemented (Damour et al., 2010; Franks et al., 2018; Sabot et al., 2022). Since the dry deposition of $O_3$ through stomata is substantial, many of these stomatal conductance models are also used to simulate the stomatal component of $O_3$ dry deposition in the atmospheric chemistry models that we analyze in this study.

Stomatal uptake of $O_3$ makes up to 45 - 75% of dry deposition during the growing season (Kurpius and Goldstein, 2003; Stella et al., 2011, 2013; Clifton et al., 2020a), with non-stomatal processes like uptake to soil and leaf cuticles and within-canopy chemistry contributing to the rest of the dry deposition. Furthermore, stomatal sensitivity to global change factors has been shown to impact tropospheric $O_3$ concentrations (Andersson and Engardt, 2010; Clifton et al., 2020b; Lin et al., 2020). For example, drought-induced declines in stomatal conductance can result in increases in $O_3$ pollution (Emberson et al., 2013; Anav et al., 2018; Clifton et al., 2020b; Lin et al., 2020). Proper representation of the stomatal component of $O_3$ $V_d$ in 3D atmospheric chemistry models is likely important for predicting $O_3$ concentrations in light of what is often referred to as the "climate penalty" on air quality, part of which includes increasing $O_3$ concentrations during drought conditions regardless of local air quality regulations (Wang et al., 2017; Lin et al., 2020).

Comparisons and evaluations of dry deposition schemes and their components, such as the stomatal component, are essential to improve the monitoring and predictive abilities of 3D atmospheric chemistry models (Dennis et al., 2010; He et al., 2021). A study by Hardacre et al. (2015) comparing $O_3$ $V_d$ and flux across global 3D atmospheric chemistry models stressed the importance of examination of the components of $O_3$ dry deposition. Wu et al. (2018) examined five dry deposition schemes at one site and suggested that the different representations of the stomatal and non-stomatal pathways contribute more to disagreements between model simulated $V_d$ compared to the different representations of the turbulent transport of $O_3$ to the land surface. Previous comparisons of dry deposition schemes also reveal that the choice of driving variables related to moisture



stress, such as near-surface VPD, leaf water potential, volumetric soil water content, or soil water potential, can be a source of disagreement between simulations of stomatal conductance (Büker et al., 2012; Visser et al., 2021; Huang et al., 2022; Sun et al., 2022; Wong et al., 2022; Clifton et al., 2023). Such evaluations often rely on observed $O_3$ flux and $V_d$ at flux towers and/or observation-based inversions of the components of $V_d$ (Fares et al., 2010; Hardacre et al., 2015; Clifton et al., 2017; Visser et al., 2021; Wong et al., 2022). Observations of latent heat flux and micrometeorological data collected over the footprint of flux towers instrumented with gas analyzers and sonic anemometers allows an inversion of the surface conductance to water vapor (Shuttleworth et al., 1984; Gerosa et al., 2007; Novick et al., 2016; Clifton et al., 2017; Medlyn et al., 2017; Knauer et al., 2018b; Vermeuel et al., 2021; Wehr and Saleska, 2021). While these estimates do not represent a direct measurement of stomatal conductance, they offer an estimate of ecosystem scale surface conductance to water vapor and the stomatal component is assumed to be dominant when transpiration dominates evapotranspiration. Such estimates of stomatal conductance have been used to understand the importance of driving variables such as $VPD$ and soil moisture for the site-scale stomatal component of $O_3$ dry deposition (e.g. Visser et al., 2021), but not to systematically evaluate many dry deposition schemes across many sites worldwide with a focus on comparing how models specify the stomatal sensitivity to driving variables. This keeps us from a full understanding of the strengths and challenges of the simulation of stomatal conductance across a large variety of dry deposition schemes. Particularly, many model parameters, within the diversity of stomatal conductance models that are used in dry deposition schemes, control the effects of moisture stress drivers on stomatal conductance, and the sensitivity of stomatal dry deposition to these parameters across a range of ecosystems is unclear.

The Air Quality Model Evaluation International Initiative 4 (AQMEII4) was designed to compare and evaluate the representation of dry deposition, including the individual contributing processes, in atmospheric chemistry models (Galmarini et al., 2021; Clifton et al., 2023). Activity 2 (hereinafter A2) isolated eighteen dry deposition schemes implemented in various atmospheric chemistry models as single-point models and forced the single-point models with micrometeorological and other environmental data from eight flux tower sites (Clifton et al., 2023). Clifton et al. (2023) used observed $O_3$ $V_d$ at the sites to evaluate modeled $O_3$ $V_d$ and carried out a detailed comparison of the components of deposition velocity. They find that simulated $O_3$ $V_d$ can be similar among models while the relative contribution of each component can be quite different among models (Clifton et al., 2023). Conversely, simulated $V_d$ can be quite variable among models when the relative contribution of the components is similar among models (Clifton et al., 2023). However, the stomatal component of $O_3$ deposition velocity simulated by the single-point models was not compared with observed $CO_2$ and latent heat flux-based estimates of stomatal conductance, which may offer observation-based insights into some of the among-model discrepancies noted by Clifton et al. (2023).

Here, we carry out a model comparison of the stomatal component of $O_3$ $V_d$ with observed flux-based estimates as a part of AQMEII4 A2. We carry out the comparison across six Northern Hemisphere sites consisting of boreal, temperate, and temperate-boreal transition forests along with an eastern Mediterranean shrubland and a temperate grassland. Particularly, we focus on parameter and process sensitivity in both model evaluation and comparison. We isolate two case studies of observed substantial decreases in soil moisture and increases in near-surface $VPD$ to understand the simulation of the stomatal component during times of increased soil and air dryness. Finally, we conducted sensitivity analyses perturbing the values of





parameters that control stomatal moisture stress to isolate the impact of parameter choice. We address the following research questions:

1. How does the stomatal component of $O_3$ $V_d$ simulated by single-point models compare with the estimates from observed latent heat and $CO_2$ flux?

2. How does the specification of moisture stress in single-point models impact the agreement in the stomatal component of $O_3$ $V_d$ between single-point model simulations and flux-based estimates during times of increased atmospheric or soil dryness?

3. In dry deposition schemes, how do stomatal moisture stress parameters affect the agreement between single-point models and observed flux-based estimates?

## 2    Methods

### 2.1    Two major classes of stomatal conductance models used in $O_3$ dry deposition schemes

Clifton et al. (2023) listed the detailed equations of all stomatal conductance models used in the single-point models evaluated in AQMEII4 A2. There are two major classes of stomatal conductance models implemented. The first class is based on the model proposed by Jarvis et al. (1976) (hereinafter Jarvis-type). Jarvis-type models use separate stress functions for different environmental conditions and specify the severity of a particular environmental stress with values which range from no stress to maximum stress. In some Jarvis-type models, individual stress functions are multiplied and serve to attenuate the maximum

stomatal conductance that can be expected under ideal growing conditions. In other Jarvis-type models, the maximum value from a set of stress functions is chosen to attenuate the maximum stomatal conductance. A range of environmental stresses are implemented among the AQMEII4 A2 dry deposition schemes that use Jarvis-type stomatal conductance models. Some schemes implement a few stress functions with incoming solar radiation and air temperature, based on the classic Wesely

(1989) dry deposition scheme, while others implement many more (Xiu and Pleim, 2001; Zhang et al., 2003). The general form of the Jarvis-type models for stomatal resistance to water vapor ($R_{s,H_2O}$; $s\ m^{-1}$) (a conductance is the inverse of a resistance) is:

$$R_{s,H_2O} = \frac{R_{s,H_2O,ideal}}{f(x_1)\ f(x_2)\ f(x_3)\ f(x_4)} \tag{1}$$

where $R_{s,H_2O,ideal}$ is the stomatal resistance to $H_2O$ under ideal environmental conditions ($s\ m^{-1}$), and $f(x)$ denotes a

function which controls the strength of the stress that is imposed by an environmental variable, $x$. Environmental variables related to air moisture include the difference between air vapor pressure and saturation vapor pressure at air temperature at measurement height ($VPD_{air}$; $kPa$), leaf-level relative humidity ($RH_l$), and relative humidity at measurement height ($RH_{z_m}$). Environmental variables related to soil moisture or plant water status include volumetric soil water content ($w_2$; $m^3\ m^{-3}$), soil matric potential ($\psi_{soil}$; $kPa$), and leaf water potential ($\psi_{leaf}$; $MPa$). A number of equations and parameters

can be used to calculate the value of $f(x)$. An example of a Jarvis-type model, implemented in the CMAQ M3Dry model (Xiu and Pleim, 2001), is:



$$R_{s,H_2O} = \frac{R_{s,H_2O,ideal}}{LAI \; f(PAR) \; f(w_2) \; f(RH_l) \; f(T_{air})} \tag{2}$$

where $PAR$ is photosynthetically active radiation ($\mu mol \; m^{-2} \; s^{-1}$), $T_{air}$ is air temperature ($°C$), and $LAI$ is leaf area index ($m^2 \; m^{-2}$).

The second class of stomatal conductance models used in the AQMEII4 A2 dry deposition schemes couple net photosynthesis with stomatal conductance using the models of Ball et al. (1987), Leuning (1995), or Medlyn et al. (2011). Net photosynthesis is simulated with the widely-used Farquhar et al. (1980) biochemical photosynthesis model or models similar to it. An example of a net photosynthesis coupled stomatal conductance model used in some of the TEMIR single-point models is (Ball et al., 1987; Tai et al., 2024):

$$R_{s,CO_2} = (\beta_t \; g_o + g_1 \frac{A_n \; RH_{z_m}}{\frac{P_{CO_2,l}}{P_a}})^{-1} \frac{P_a}{R\theta_a} \tag{3}$$

where $R_{s,CO_2}$ is the stomatal resistance to $CO_2$ ($s \; m^{-1}$), $g_o$ is the minimum conductance ($mol \; m^{-2} \; s^{-1}$), $g_1$ is a slope parameter commonly used in the models of Leuning (1995), Ball et al. (1987), and Medlyn et al. (2011) (the interpretation of $g_1$ is different among models), $A_n$ is net photosynthesis ($mol \; m^{-2} \; s^{-1}$), $P_{CO_2,l}$ is $CO_2$ partial pressure at the leaf surface ($Pa$), $P_a$ is the air pressure ($Pa$), $R$ is the universal gas constant ($J \; mol^{-1} \; K^{-1}$), and $\theta_a$ is potential temperature ($K$). The

soil moisture stress factor, $\beta_t$, is calculated as:

$$\beta_t = \begin{cases} 1, & \psi_{soil} > \psi_{soil,fc} \\ \frac{\psi_{soil,wlt} - \psi_{soil}}{\psi_{soil,wlt} - \psi_{soil,fc}}, & \psi_{soil,wlt} \le \psi_{soil} \le \psi_{soil,fc} \\ 0, & \psi_{soil} < \psi_{soil,fc} \end{cases} \tag{4}$$

where $\psi_{soil,fc}$ is the soil matric potential at field capacity ($kPa$), and $\psi_{soil,wlt}$ is the soil matric potential at wilting point ($kPa$). Like the Jarvis-type models, participating net photosynthesis models vary in whether they employ $RH_{z_m}$, $RH_l$ or $VPD_{air}$ to incorporate the effects of air moisture on stomatal conductance. Soil moisture impacts are simulated with the use

of $\psi_{soil}$ or $w_2$ to calculate stress factors (e.g. $\beta_t$) among the net photosynthesis coupled models. In models that use $\psi_{soil}$ in their soil moisture stress function, $\psi_{soil}$ is estimated as a function of $w_2$ as:

$$\psi_{soil} = \psi_{soil,sat} w_2^{-B} \tag{5}$$

where $\psi_{soil,sat}$ is the soil matric potential at saturation ($kPa$). $R_{s,H_2O}$ is scaled to the stomatal resistance to $O_3$ using the ratio of $O_3$ diffusivity in air ($m^2 \; s^{-1}$) to $H_2O$ diffusivity in air ($m^2 \; s^{-1}$). $R_{s,CO_2}$ is scaled to the stomatal resistance to $O_3$

using the ratio of $O_3$ diffusivity in air ($m^2 \; s^{-1}$) to $CO_2$ diffusivity in air ($m^2 \; s^{-1}$).



Both the net photosynthesis coupled models and the Jarvis-type models have a number of parameters that control the effect of soil and/or air moisture on stomatal conductance. Full descriptions of the single-point models in this study can be found in (Clifton et al., 2023). We divide the two major classes of stomatal conductance models into 4 classes as: net photosynthesis coupled models that include both soil and air moisture impacts through the use of $\psi_{soil}$, $w_2$, $VPD_{air}$, $RH_{z_m}$, or $RH_l$

(NP:SM/VPD/RH), Jarvis-type models that include both soil and air moisture impacts through the use of $w_2$, $VPD_{air}$, $RH_{z_m}$, or $RH_l$ (J:SM/VPD/RH), Jarvis-type models that only include $VPD_{air}$ (J:VPD), and Jarvis-type models that do not include soil moisture or air moisture impacts (J:NoSM/VPD/RH). J:VPD models include the impact of plant water status through the use of $\psi_{leaf}$. However, $\psi_{leaf}$ is only modeled as a function of solar radiation, and it is not coupled with $\psi_{soil}$. Therefore, we did not include it in the J:SM/VPD/RH class.

**2.2 Stomatal conductance to O₃ estimated from observations of latent heat flux and net ecosystem exchange of CO₂**

We calculated two separate estimates of stomatal conductance to $O_3$ ($G_{s,O_3}$) using observations of latent heat flux ($\lambda E$) and net ecosystem exchange of $CO_2$ ($NEE$) at the flux towers. These flux observations are not used as forcing data for the single point models. The first estimate is an inversion of the evaporation-resistance form of the Penman-Monteith (PM) equation (Monteith, 1981) as presented by Gerosa et al. (2005, 2007) to calculate the surface resistance to water vapor, $R_{s,H_2O,PM}$ ($s\ m^{-1}$), as:

$$R_{s,H_2O,PM} = \frac{\rho c_p [VPD_{canopy-air}]}{\gamma \lambda E} - (R_a + R_{b,H_2O}) \tag{6}$$

where $\lambda E$ is the latent heat flux ($W\ m^{-2}$), $\rho$ is the air density ($kg\ m^{-3}$), $c_p$ is the specific heat capacity of dry air at constant pressure ($J\ K^{-1}\ kg^{-1}$), $VPD_{canopy-air}$ is the vapor pressure difference between the evaporating canopy surface and the measurement height ($kPa$), $\gamma$ is the psychrometric constant ($kPa\ K^{-1}$), $R_a$ is the aerodynamic resistance to turbulent transfer ($s\ m^{-1}$), and $R_{b,H_2O}$ is the quasi-laminar layer resistance to water vapor ($s\ m^{-1}$). $VPD_{canopy-air}$ is calculated as:

$$VPD_{canopy-air} = e_s(T_s) - e(z_m) \tag{7}$$

where $e_s(T_s)$ is the saturation vapor pressure ($kPa$) at the temperature of the evaporating canopy surface ($T_s$; $°C$), and $e(z_m)$ is the vapor pressure at the measurement height ($kPa$). $e_s(T_s)$ is calculated using Tetens formula:

$$e_s(T_s) = a\ exp(\frac{bT_s}{T_s + c}) \tag{8}$$

where $a = 0.611\ kPa$, $b = 17.502$, and $c = 240.97\ °C$. $T_s$ was calculated as:

$$T_s = T_{air} + \frac{H}{\rho c_p}(R_a + R_{b,H}) \tag{9}$$

where $H$ is the sensible heat flux ($W\ m^{-2}$), and $R_{b,H}$ is the quasi-laminar layer resistance to heat ($s\ m^{-1}$).





The second stomatal conductance estimate fits a stomatal conductance model to the estimates of stomatal conductance from the PM inversion ($G_{s,H_2O,PM} = R^{-1}_{s,H_2O,PM}$). We fit the Medlyn et al. (2011) leaf-level stomatal conductance model using gross primary productivity ($GPP$) as presented by Medlyn et al. (2017) and Knauer et al. (2018b):

$$G_{s,H_2O,MED} = G_o + 1.6 \, (1 + \frac{G_1}{\sqrt{VPD_{canopy-air}}}) \, \frac{GPP}{C_a} \tag{10}$$

where $G_{s,H_2O,MED}$ is the stomatal conductance to H$_2$O ($mol \ m^{-2} \ s^{-1}$), $G_o$ is the minimum stomatal conductance to H$_2$O ($mol \ m^{-2} \ s^{-1}$), $G_1$ is a parameter that is inversely related to intrinsic water use efficiency ($kPa^{0.5}$) (Medlyn et al., 2017), $GPP$ is gross primary productivity ($\mu mol \ m^{-2} \ s^{-1}$), and $C_a$ is the ambient CO$_2$ concentration ($ppm$). $GPP$ was calculated by partitioning the $NEE$ flux into $GPP$ and ecosystem respiration, $R_{eco}$. Nighttime $NEE$ is assumed to be entirely $R_{eco}$.

$NEE$ was partitioned using the R package REddyProc with a partitioning approach that uses nighttime $NEE$ flux to estimate a temporally varying $R_{eco}$-$T_{air}$ relationship as (Reichstein et al., 2012; Wutzler et al., 2018; Stoy et al., 2006):

$$R_{eco}(T_{air}) = R_{Ref} \, exp[E_0(\frac{1}{T_{air,Ref} - T_{air,0}} - \frac{1}{T_{air} - T_{air,0}})] \tag{11}$$

where $E_0$ is the temperature sensitivity, $T_{air,0}$ is held constant at $-46.02 \ °C$, and $T_{air,Ref}$ is held at $15 \ °C$. The $R_{eco}$-$T_{air}$ relationship is applied to daytime data to obtain estimates of $R_{eco}$ during the day. Finally, $R_{eco}$ estimates are used to estimate

$GPP$ as:

$$GPP = R_{eco} - NEE \tag{12}$$

In order to estimate the $G_o$ and $G_1$ parameters by fitting equation 10 to $G_{s,H_2O,PM}$, we limited the full data that was used for the AQMEII4 A2 to conditions when transpiration would dominate $\lambda E$ and the stomatal component would dominate the surface conductance to water vapor. To do this, we limited the data to daytime conditions when relative humidity is less than

80%. Data collected during a precipitation event and 48 hours after a precipitation event was removed. We also limited the data to conditions when sensible heat flux was positive to avoid stable atmospheric conditions. Finally, we removed unusually high values for $G_{s,H_2O,PM}$ determined by looking at growing season daily box plots of $G_{s,H_2O,PM}$ at each site. The site-specific upper thresholds for $G_{s,H_2O,PM}$ are listed in Table 1. Table 1 also shows other flux tower site details. The Python-based open-source software, SciPy, was used to estimate $G_o$ and $G_1$ through least squares optimization using a Huber loss function to

reduce the influence of outliers (Virtanen et al., 2020). Calculating $G_{s,H_2O,MED}$ and $G_{s,H_2O,PM}$ allowed us to use the $GPP$ flux which is partitioned from $NEE$ calculated from observed CO$_2$ flux. This adds an estimate of stomatal conductance which is driven by $GPP$ in addition to a separate estimate which is driven by $\lambda E$. Both CO$_2$ flux and $\lambda E$ share the stomatal pathway in their total flux. In order to calculate $G_{s,O_3,PM}$ and $G_{s,O_3,MED}$, $G_{s,H_2O,PM}$ and $G_{s,H_2O,MED}$ were scaled by the ratio of O$_3$ diffusivity ($D_{O_3}$) and H$_2$O diffusivity ($D_{H_2O}$). The ratio, $\frac{D_{O_3}}{D_{H_2O}}$, was set as 0.61. All of the analysis in this paper uses

daytime flux tower data and single point model simulations.





### 2.2.1 Effective stomatal conductance to O$_3$

The effective stomatal conductance to O$_3$ ($eg_s$) quantifies the amount of O$_3$ $V_d$ that can be attributed to the stomatal component, and it is used to compare the contribution of a given depositional pathway (e.g. stomatal uptake) across models with differences in resistance schemes (Paulot et al., 2018; Clifton et al., 2020b; Galmarini et al., 2021). The single-point model simulations of

$eg_s$ are archived with AQMEII4 A2. It is important to note that the exact calculation of the effective conductance for a given pathway from the single-point models depends on the resistance framework used (Galmarini et al., 2021).

To compare $eg_s$ from the single-point models and the observed flux-based estimates, we calculated $eg_s$ from the (half-)hourly observed flux-based estimates, $G_{s,O_3,PM}$ and $G_{s,O_3,MED}$, as:

$$eg_{s,PM} = \frac{G_{s,O_3,PM}}{G_c} V_d \qquad (13)$$

$$eg_{s,MED} = \frac{G_{s,O_3,MED}}{G_c} V_d \qquad (14)$$

where $G_c$ is the canopy conductance to O$_3$ ($1/R_{c,O_3}$). $R_{c,O_3}$ is calculated as the residual in $V_d^{-1}$ after calculating $R_a$ and the quasi-laminar boundary layer resistance to O$_3$ ($R_{b,O_3}$; $s\ m^{-1}$) using the following big-leaf resistance framework:

$$R_{c,O_3} = V_d^{-1} - R_a - R_{b,O_3} \qquad (15)$$

$R_a$ was calculated as (Verma, 1989; Knauer et al., 2018a):

$$R_a = \frac{u(z_m)}{u*^2} \qquad (16)$$

where $u(z_m)$ is the wind speed ($m\ s^{-1}$) at the measurement height and $u*$ is the friction velocity ($m\ s^{-1}$). $R_{b,O_3}$ is calculated as:

$$R_{b,O_3} = \frac{2}{ku*} \left(\frac{Sc}{Pr}\right)^{2/3} \qquad (17)$$

where $k$ is the von Karman constant (0.4), $Sc$ is the Schmidt number (the ratio of kinematic viscosity of air to the molecular

diffusivity of O$_3$) and $Pr$ is the Prandtl number (the ratio of kinematic viscosity to thermal diffusivity).

To answer question 1, we calculated monthly mean $eg_s$ from each single-point model and compared the averages to the monthly mean $eg_{s,PM}$ and $eg_{s,MED}$ inferred from observations as was done for total O$_3$ $V_d$ in Clifton et al. (2023). We calculated monthly averages using all available years at a site. Therefore, when multiple years of data is available for a month, the monthly mean is a multiyear monthly mean. Ramat Hanadiv and Bugacpuszta only have one year of data for certain months





**Table 1.** Description of the observational data used at each flux tower.

| Site | Location | Temporal resolution | Site years used [a] | $G_{s,H_2O,PM}$ UL [b,c] |
|---|---|---|---|---|
| Harvard Forest, USA | 42.54° N, 72.17° W | Hourly | 1992 - 2000 | $0.03\ m\ s^{-1}$ |
| Borden Forest, Canada | 44.32° N, 79.93° W | Half-hourly | 2008 - 2013 | $0.03\ m\ s^{-1}$ |
| Ispra, Italy | 45.81° N, 8.63° E | Half-hourly | 2013 - 2015 | $0.02\ m\ s^{-1}$ |
| Hyytiälä, Finland | 61.85° N, 24.29° E | Half-hourly | 2002 - 2005, 2007 - 2012 | $0.02\ m\ s^{-1}$ |
| Ramat Hanadiv, Israel | 32.55° N, 34.93° E | Half-hourly | 2016, 2017 | $0.02\ m\ s^{-1}$ |
| Bugacpuszta, Hungary | 46.69° N, 19.60° E | Half-hourly | 2012, 2013 | $0.02\ m\ s^{-1}$ |

[a] Figure S1 displays the months with available observations during each year used.

[b] $G_{s,H_2O,PM}$ is the stomatal conductance to $H_2O$ using the Penman-Monteith inversion.

[c] $G_{s,H_2O,PM}$ UL stands for the upper limit applied to $G_{s,H_2O,PM}$. A description of the selection of the upper limit is provided in section 2.2.

of the year (Figure S1). We also compared the minimum, maximum, and central range of single-point modeled $eg_s$ monthly averages with monthly mean $eg_{s,PM}$ and $eg_{s,MED}$ as Clifton et al. (2023) introduced for evaluating total $O_3$ $V_d$ from these single-point models. We calculated the interquartile range (IQR) of the monthly averages from all single-point models, and we call the IQR the "central range" throughout the paper.

## 2.3 Isolating times of water stress for case studies

We chose two case studies to address question 2 by comparing the agreement between single-point modeled $eg_s$ and observed flux-based $eg_s$ during times of water stress: Borden Forest and Ramat Hanadiv. These case studies were chosen because they represent times when there was substantial disagreement in $eg_s$ among models and between models and observed flux-based estimates, $eg_{s,PM}$ and $eg_{s,MED}$, and the specification of moisture stress appeared to be a source of the variability. The first case study is at Borden Forest, Canada. The monthly mean soil volumetric water content measured at 50 cm depth fell below 240 the model specified wilting point during July 2011 and 2012 and September 2009, 2010, and 2011 at Borden Forest. July 2011 and 2012 also exhibit high mean $VPD_{air}$ along with low mean soil volumetric water content (Figure S1). Therefore, we used July 2011 and 2012 at Borden Forest as the first case study and compared simulated $eg_s$ with $eg_{s,PM}$ and $eg_{s,MED}$ to understand if the flux-based $eg_s$ supports the single-point modeled $eg_s$ during this time. The second case study is at Ramat Hanadiv, Israel, a seasonally dry shrubland which experiences sharp declines in soil moisture and increases in $VPD_{air}$ during 245 the dry summer months (Figure S1). Clifton et al. (2023) showed large divergence between single-point modeled and observed $V_d$ at Ramat Hanadiv during the dry summer months. The seasonally dry months provided a case where we were able to study if models capture $eg_s$ during conditions when the vegetation experiences substantial water stress at this shrubland.





## 2.4 Sensitivity of single-point simulated $eg_s$ to parameters that control stomatal moisture stress

Sensitivity simulations were conducted to isolate the impact of moisture stress parameters on single-point modeled $eg_s$. The
parameters related to moisture stress from many participating dry deposition schemes were perturbed along a range of values
listed in Table 2. Some parameters control the strength of the soil moisture stress and the strength of the $VPD_{air}$ stress. Other
parameters control other plant ecophysiological properties such as the relationship between net photosynthesis and stomatal
conductance or the intrinsic water-use efficiency. Additionally, we perturbed the $R_{s,H_2O,ideal}$ and $G_{s,H_2O,max}$ parameters
as J:NoSM/VPD/RH models do not include moisture stress variables. Lastly, $VPD_{air}$ and $RH_l$ stress functions in some
J:SM/VPD/RH models did not have parameters. For these models, we perturbed the value of the stress functions, $f(VPD)$
and $f(RH_l)$ to test the impact of varying the strength of the $VPD_{air}$ and $RH_l$ stress.

A set of sensitivity simulations consisted of 5 - 7 simulations conducted for a given parameter or stress function in which
the value of the parameter or stress function was changed within the range listed in Table 2. In total, 12 parameters and 2 stress
functions without parameters were investigated. The number of sets of sensitivity simulations and the number of sensitivity
simulations within a set are different across models for a couple of reasons. First, stomatal conductance models vary in the
driving variables they use for moisture stress, and certain parameters are unique to a given model. Second, while many stomatal
conductance models share parameters, different model implementations mean that the parameters need to be perturbed across
different ranges to capture the effect of that parameter on $eg_s$ for a given model.

We conducted the sensitivity simulations to answer question 3 and understand how moisture stress parameter values impact
the agreement between single-point modeled $eg_s$ and flux-based $eg_s$ for our two case studies at Borden Forest and Ramat
Hanadiv as well as other sites. We calculated the median absolute difference ($MAD$) between single-point modeled $eg_s$ and
flux-based $eg_s$ as

$$MAD_{MED,m,p,v} = Median \left| eg_{s,MED} - eg_{s,SPMod_{m,p,v}} \right| \tag{18}$$

$$MAD_{PM,m,p,v} = Median \left| eg_{s,PM} - eg_{s,SPMod_{m,p,v}} \right| \tag{19}$$

where $eg_{s,SPMod_{m,p,v}}$ is the estimate of $eg_s$ from a single-point model for model $m$ in $1,...,M$, parameter or stress func-
tion $p$ in $1,...,P$, and parameter or stress function value $v$ in $1,...,V$. We calculated one summertime $MAD_{MED,m,p,v}$ and
$MAD_{PM,m,p,v}$ for Borden Forest, Harvard Forest, Hyytiälä, and Ispra by pooling the (half-)hourly absolute differences for
June, July, and August. We calculated three $MAD_{MED,m,p,v}$ and $MAD_{PM,m,p,v}$ for Ramat Hanadiv by pooling together
the absolute differences for winter, spring, and summer separately. At Ramat Hanadiv, we pooled the (half-)hourly abso-
lute differences in January and February for the winter $MAD_{MED,m,p,v}$ and $MAD_{PM,m,p,v}$, March - April for the spring
$MAD_{MED,m,p,v}$ and $MAD_{PM,m,p,v}$, and June - September for the summer $MAD_{MED,m,p,v}$ and $MAD_{PM,m,p,v}$. For each
parameter or stress function in a model, we calculated the change in $MAD_{MED,m,p,v}$ and $MAD_{PM,m,p,v}$ with change in the
parameter or stress function value as:





$$\frac{\Delta MAD_{MED,m,p}}{\Delta v_{m,p}} = \frac{MAD_{MED,m,p,v,Max} - MAD_{MED,m,p,v,Min}}{v_{m,p,Max} - v_{m,p,Min}} \tag{20}$$

$$\frac{\Delta MAD_{PM,m,p}}{\Delta v_{m,p}} = \frac{MAD_{PM,m,p,v,Max} - MAD_{PM,m,p,v,in}}{v_{m,p,Max} - v_{m,p,Min}} \tag{21}$$

where $MAD_{MED,m,p,v,Max}$ is the maximum $MAD_{MED,m,p,v}$, $MAD_{MED,m,p,v,Min}$ is the minimum $MAD_{MED,m,p,v}$, $MAD_{PM,m,p,v,Max}$ is the maximum $MAD_{PM,m,p,v}$, $MAD_{PM,m,p,v,Min}$ is the minimum $MAD_{PM,m,p,v}$, $v_{m,p,Max}$ is the maximum parameter or stress function value, and $v_{m,p,Min}$ is the minimum parameter or stress function value. One of the $W_{wlt}$, $\psi_{soil,wlt}$, and $\psi_{leaf,min}$ sensitivity simulations perturbed the parameter value to an extremely low value, -1E+09, to understand the impact of substantially lowering the strength of the soil moisture stress for the Borden Forest case study. However, we did not use this sensitivity simulation in calculating $\frac{\Delta MAD_{MED,m,p}}{\Delta v_{m,p}}$ for these parameters to avoid a large $\Delta v_{m,p}$. We will refer to $\frac{\Delta MAD_{PM,m,p}}{\Delta v_{m,p}}$ and $\frac{\Delta MAD_{MED,m,p}}{\Delta v_{m,p}}$ as simply $\frac{\Delta MAD_{PM}}{\Delta v}$ and $\frac{\Delta MAD_{MED}}{\Delta v}$, respectively, throughout the remaining discussion.

To study the impact of parameter perturbations on model bias specifically for our case studies that suggested water stress related over- or underestimation of $eg_s$, we calculated the monthly median difference between single-point modeled $eg_s$ and the two flux-based $eg_s$ estimates, $eg_{s,PM}$ and $eg_{s,MED}$ for both base and sensitivity simulations as:

$$MD_{eg_s,MED,m,p,v} = Median\left(eg_{s,MED} - eg_{s,SPMod_{m,p,v}}\right) \tag{22}$$

$$MD_{eg_s,PM,m,p,v} = Median\left(eg_{s,PM} - eg_{s,SPMod_{m,p,v}}\right) \tag{23}$$

For Borden Forest, we calculated the monthly median $MD_{eg_s,MED,m,p,v}$ and $MD_{eg_s,PM,m,p,v}$ only using (half-)houlry differences during 2011 and 2012 to focus on the years of our case study. For Ramat Hanadiv, we calculated monthly median $MD_{eg_s,MED,m,p,v}$ and $MD_{eg_s,PM,m,p,v}$ using all available years. Since the median differences for the case studies are calculated for the base simulation as well, $1, ..., V$ includes the parameter or stress function value used in the base simulation. We will refer to $MD_{eg_s,MED,m,p,v}$ and $MD_{eg_s,PM,m,p,v}$ as simply $MD_{eg_s,MED}$ and $MD_{eg_s,PM}$ hereinafter. A negative value of $MD_{eg_s,MED}$ and $MD_{eg_s,PM}$ means overestimation of $eg_s$ by the single point model and a positive value means underestimation by the single point model relative to the flux-based $eg_s$.

## 3 Results

### 3.1 Monthly averages of single-point modeled $eg_s$ across sites and comparison with observed flux-based estimates of $eg_s$

We first compare monthly averages from the two different observed flux-based estimates, $eg_{s,PM}$ and $eg_{s,MED}$, which are denoted as "Flux-based: $eg_{s,PM}$" and "Flux-based: $eg_{s,MED}$" respectively in Figure 1. We find that the seasonal cycle of $eg_s$





**Figure 1.** Monthly daytime averages of flux-based $eg_s$ marked as Flux-based: $eg_{s,PM}$ and Flux-based: $eg_{s,MED}$ and single-point modeled $eg_s$. Column 1 shows the model central range, minimum, and maximum of monthly daytime averages of all single-point modeled $eg_s$. Column 2 - 5 show monthly daytime averages of single-point modeled $eg_s$ by stomatal conductance model type. NP is net photosynthesis coupled. J is Jarvis-type. SM is soil moisture, $VPD$ is near-surface $VPD$, and RH can be relative humidity at leaf surface or measurement height. Rows are labeled by site. Dots show the monthly mean $eg_s$ and vertical bars show $\pm$ 2 standard error of the mean.





**Table 2.** A list of parameters related to moisture stress used in the dry deposition schemes. The "Values" column lists the range of the values used in the sensitivity analysis during which the values of a given parameter were changed to values within the range listed.

| Parameter Group [a] | Parameter | Values | $G_s$ model type [b,c] | Models |
|---|---|---|---|---|
| Initial resistance or conductance | $R_{s,H_2O,ideal}$ $(s\ m^{-1})$ | [100,250] | J:SM/VPD/RH, J:VPD, J:NoSM/VPD/RH | WRF-Chem Wesely, GEOS-Chem Wesely |
| | | | | IFS SUMO Wesely, IFS GEOS-Chem Wesely |
| | | | | GEM-MACH Wesely, GEM-MACH Zhang, CMAQ STAGE |
| | | | | TEMIR Wesely, TEMIR Zhang |
| | $G_{s,H_2O,max}$ $(m\ s^{-1})$ | [0.001, 0.007] | J:SM/VPD/RH | DO$_3$SE multi |
| $VPD$ stress | $VPD_{max}$ $(kPa)$ | [2.5, 0.5] | J:SM/VPD/RH | DO$_3$SE multi |
| | $B_{VPD}$ $(kPa^{-1})$ | [0, 0.5] | J:VPD | GEM-MACH Zhang, TEMIR Zhang |
| | $D_o$ $(kPa)$ | [2, 10] | NP:SM/VPD/RH | DO$_3$SE psn |
| | $f(VPD)$ | [0, 1] | J:SM/VPD/RH, J:VPD | IFS SUMO Wesely, GEM-MACH Wesely |
| $RH$ stress | $f(RH_l)$ | [0, 1] | J:SM/VPD/RH | CMAQ STAGE |
| Soil moisture stress | $W_{wlt}$ $(m^3\ m^{-3})$ | [-1E+09, 0.1] | J:SM/VPD/RH, NP:SM/VPD/RH | IFS SUMO Wesely, CMAQ STAGE |
| | | | | DO$_3$SE multi, DO$_3$SE psn, MLC-CHEM |
| | $\psi_{leaf,min}$ $(MPa)$ | [-1E+09, -500] | J:VPD | GEM-MACH Zhang, TEMIR Zhang |
| | $\psi_{soil,wlt}$ $(kPa)$ | [-1E+09, -2.75] | NP:SM/VPD/RH | TEMIR Wesely BB, TEMIR Wesely Medlyn |
| | $B$ | [2.5, 7] | NP:SM/VPD/RH | TEMIR Wesely BB, TEMIR Wesely Medlyn |
| Slope controlling the relationship between $G_s$ and $A_n$ | $g_{1,L}$ | [6, 11] | NP:SM/VPD/RH | DO$_3$SE psn, MLC-CHEM |
| | $g_{1,BB}$ | [4, 11] | NP:SM/VPD/RH | TEMIR Wesely BB |
| | $g_{1,M}$ | [1.5, 5.5] | NP:SM/VPD/RH | TEMIR Wesely Medlyn |

[a] $G_s$ is stomatal conductance. $A_n$ is net photosynthesis. $VPD$ is $VPD_{air}$. $RH$ can be $RH_{z_m}$ or $RH_l$ depending on singe-point model.

[b] $G_s$ is stomatal conductance.

[c] NP:SM/VPD/RH models are net photosynthesis coupled models that include $\psi_{soil}$, $w_2$, $VPD_{air}$, $RH_{z_m}$, or $RH_l$ impacts of stomatal conductance. J:SM/VPD/RH models are Jarvis-type models that include $w_2$, $VPD_{air}$, $RH_{z_m}$, or $RH_l$ impacts on stomatal conductance. J:VPD models are Jarvis-type models that only include $VPD_{air}$ impacts on stomatal conductance. J:NoSM/VPD/RH models are Jarvis-type models that do not include soil moisture or air moisture impacts.

from the two estimates agree at most sites, but the magnitude of $eg_s$ diverged at all sites at some point during the growing season (Figure 1). The largest disagreements between the two observed flux-based estimates occurred at Borden Forest and Ramat Hanadiv. At Borden Forest, $eg_{s,PM}$ was higher than $eg_{s,MED}$ from April to October. At Ramat Hanadiv, $eg_{s,PM}$ was higher than $eg_{s,MED}$ during the winter months and lower than $eg_{s,MED}$ during the spring and summer months. These disagreements at Ramat Hanadiv and Borden Forest can be partly because stomatal conductance estimates from the PM inversion, $G_{s,H_2O,PM}$,

are higher than those from $G_{s,H_2O,MED}$ during times of low $VPD_{canopy-air}$. Furthermore, when looking at the relationship between the stomatal conductance estimate and the underlying flux used in the estimate, $G_{s,H_2O,MED}$ is more tightly coupled with $NEE$ and the partitioned $GPP$ compared to the coupling between $G_{s,H_2O,PM}$ an $\lambda E$ at all sites.

We next compare the observed flux-based estimates, $eg_{s,PM}$ and $eg_{s,MED}$, to the single-point models. In addition to the observed flux-based estimates, Figure 1 shows the $eg_s$ estimates from each single-point model as divided into the four classes

of stomatal conductance models described in section 2.1. Figure 1 also displays the central range, the maximum value, and the minimum values of the monthly averages from all single-point models. The seasonal cycle of $eg_{s,PM}$ and $eg_{s,MED}$ agrees with the seasonal cycle of the central range at most forest sites (Figure 1). Specifically, monthly mean $eg_{s,MED}$ and $eg_{s,PM}$ estimates fall within the central range (Figure 1) throughout times of peak $GPP$ at Harvard Forest, Borden Forest, and Ispra during June, July, and August (Figure S1). At Harvard Forest, Borden Forest, and Ispra, the central range suggests an increase

in $eg_s$ into the summer months (June, July, August) with declines after September. Monthly averages for $eg_{s,PM}$ and $eg_{s,MED}$ display the same seasonal cycle at these forest sites. Even though the seasonal cycle of $eg_s$ suggested by the central range





is supported by the flux-based averages, individual model averages can disagree with the $eg_{s,PM}$ and $eg_{s,MED}$ averages at these three forests (Figure 1). For example, GEM-MACH Wesely averages show an earlier and more abrupt decline in $eg_s$ into September after the summer months at Harvard Forest, Borden Forest, and Ispra compared to the averages from some other single-point models, $eg_{s,PM}$, and $eg_{s,MED}$.

At the boreal forest in Hyytiälä, the April - July increase in $eg_s$ in the single-point model central range is well supported by the $eg_{s,MED}$ and $eg_{s,PM}$ averages (Figure 1). However, the agreement between the central range and flux-based averages degrades past July. The flux-based estimates continue to show increasing $eg_s$ into August while the upper limit of the central range begins to decline past the July peak (Figure 1). Among the forests, the largest disagreement between flux-based $eg_s$ and the central range occurred at Hyytiälä during times of peak $GPP$ in August and September (Figure 1 and Figure S1). During these months, the multiyear mean $eg_{s,PM}$ and $eg_{s,MED}$ are higher compared to the central range and most single-point models. GEM-MACH Wesely averages show a more rapid decline in $eg_s$ past the June peak compared to the other single-point models at Hyytiälä. Furthermore, IFS SUMO Wesely averages suggest declining $eg_s$ from March to December while most other models do not show post peak $eg_s$ declines until after July (Figure 1).

At the shrubland site, Ramat Hanadiv, the monthly mean $eg_{s,MED}$ falls within the model central range during the winter and spring months (January - May) (Figure 1). The central range and both $eg_{s,MED}$ and $eg_{s,PM}$ averages suggest declining $eg_s$ into the dry summer months at this site. However, the central range suggests a more flat seasonal cycle in $eg_s$ with less variability between the wet months and dry months compared to $eg_{s,MED}$ and $eg_{s,PM}$. Compared to the forests examined here, there is stronger disagreement about the seasonal cycle of $eg_s$ among individual single-point models at Ramat Hanadiv. For example, some models, such as GEM-MACH Zhang, TEMIR Zhang, GEM-MACH Wesely, WRF-Chem Wesely, and CMAQ STAGE estimate higher $eg_s$ during the dry summer months compared to the wet winter months, which is not supported by $eg_{s,MED}$ and $eg_{s,PM}$ (Figure 1). Some single-point models like those from the TEMIR models, GEM-MACH Wesely, and WRF-Chem Wesely show rapidly increasing $eg_s$ into the spring months (March - May), which is not supported by $eg_{s,MED}$ or $eg_{s,PM}$ averages. Other models like CMAQ M3Dry models, IFS SUMO Wesely, TEMIR Wesely, and IFS GEOS-Chem Wesely show relatively less month-to-month variability in $eg_s$ compared to $eg_{s,MED}$ and $eg_{s,PM}$ (Figure 1).

Out of all of the ecosystems studied here, the highest disagreement in monthly means between modeled and flux-based $eg_s$ occurs at Bugacpuszta where there is unfortunately missing data during June and July. Clifton et al. (2023) also noted this for $O_3$ $V_d$ at this site. Many single-point models as well as the central range show a single and sharply peaked seasonal cycle with a maximum during June and July (Figure 1). The limited observations at Bugacpuszta used for this activity do not allow us to confidently estimate the full seasonal cycle of flux-based $eg_s$ at this site. Looking at the months when observations were available, $eg_{s,MED}$ and $eg_{s,PM}$ are higher than the central range and the maximum modeled monthly averages during most available months with the exception of May, August, and September (Figure 1). Thus, neither the central range nor individual single-point models simulate $eg_s$ monthly averages that are in agreement with flux-based estimates. As Clifton et al. (2023) noted, during August and September, soil volumetric water content was below model specified wilting point (Figure S1) and many single-point models that include soil moisture stress simulate very low $eg_s$ lowering the central range. $eg_{s,MED}$ and $eg_{s,PM}$ also show lowered $eg_s$ during the limited observations in August and September.





## 3.2 Comparison of $eg_s$ during moisture stress: Case studies at Borden Forest and Ramat Hanadiv

Figure 2 shows a comparison between single-point modeled and observed flux-based $eg_s$ during times of observed increases in soil and atmospheric dryness. The top panel shows the comparison for Borden Forest and the bottom panel shows the comparison for Ramat Hanadiv. For Borden Forest, the purple box plot shows the distribution of (half-)hourly July $eg_s$ during 2011 and 2012. The green box plot shows the distribution of (half-)hourly July $eg_s$ during all years excluding 2011 and 2012. The flux-based estimates of $eg_s$, $eg_{s,PM}$ and $eg_{s,MED}$, demonstrate that the observed decreases in soil moisture and increases in $VPD_{air}$ during July 2011 and 2012 did not lower stomatal conductance or $eg_s$ at Borden Forest (Figure 2). Similarly, some single-point models do not show reductions in $eg_s$ during July 2011 and 2012 compared to the July distribution of $eg_s$ outside of 2011 and 2012 (Figure 2). However, other single-point models show too large reductions in $eg_s$ during July 2011 and 2012 (Figure 2); these are models that use specific functions to simulate the effect of soil moisture on stomatal conductance (the NP:SM/VPD/RH and J:SM/VPD/RH models). Thus, single-point models that simulate the effect of soil moisture on stomatal conductance underestimated $eg_s$ compared to flux-based estimates at this temperate-boreal transition forest.

For Ramat Hanadiv, the box plots labeled "Winter" show the distribution of (half-)hourly January - February $eg_s$, the box plots labeled "Spring" show the distribution of (half-)hourly March - May $eg_s$, and the box plots labeled "Summer" show the distribution of (half-)hourly June - September $eg_s$. Ramat Hanadiv experienced periods of observed decreases in soil moisture and increases in $VPD_{air}$ during the summer months, and the shrubland does respond with decreases in observed flux-based $eg_s$ (Figure 2). As discussed in Section 3.2, individual models vary greatly in their seasonal cycles of $eg_s$ at this site. For example, TEMIR Zhang, GEM-MACH Zhang, GEM-MACH Wesely, WRF-Chem Wesely, and CMAQ STAGE simulate the opposite seasonal cycle in $eg_s$ compared to $eg_{s,PM}$ and $eg_{s,MED}$, with higher $eg_s$ during the dry summer months compared to the wet winter months (Figure 2). These models that do not capture the observed seasonality in $eg_s$ or suggest peak $eg_s$ during the dry months at Ramat Hanadiv are Jarvis-type models that do not include specific stress functions for soil moisture stress (J:VPD or J:NoSM/VPD/RH) with the exception of CMAQ STAGE. TEMIR Zhang and GEM-MACH Zhang use a leaf water potential function to simulate moisture stress, and we find that the simulated effect is not in agreement with flux-based estimates for this shrubland. GEM-MACH Wesely and WRF-Chem Wesely use season specific initial resistances, $R_{s,H_2O,ideal}$, set to very large values for winter and autumn which is in disagreement with the seasonal cycle of flux-based stomatal conductance at this site.

The two cases presented here reveal that current model formulations of the effects of moisture stress on stomatal conductance can both over- and underestimate these effects relative to what is implied by flux-based stomatal conductance estimates. The Borden Forest case represents a case where $eg_s$ was underestimated by many models compared to $eg_{s,PM}$ and $eg_{s,MED}$ when observed soil moisture fell below model threshold determined by parameter choice. The Ramat Hanadiv case represents a case where $eg_s$ was overestimated by many models compared to $eg_{s,PM}$ and $eg_{s,MED}$ during the dry summer months. Both simulated moisture stress and the seasonality of stomatal conductance through the specification of season-specific initial resistances appears to contribute to disagreement in $eg_s$. In the next section, we focus on the sensitivity of the agreement between single-point modeled and flux-based $eg_s$ to changes in the values of model parameters that control: stomatal moisture stress,



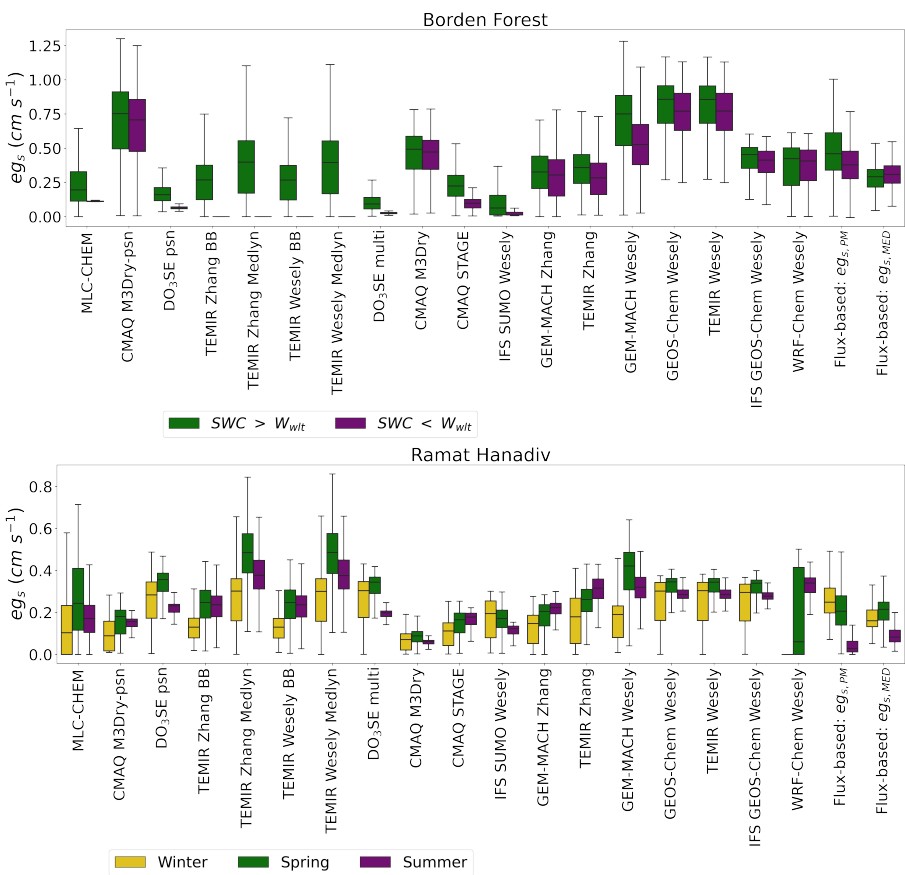

**Figure 2.** Comparisons of simulated $eg_s$ and observed flux-based $eg_s$, "Flux-based: $eg_{s,PM}$" and "Flux-based: $eg_{s,MED}$", for sites used in two case studies. Top panel shows box plots of Borden Forest, daytime, (half-)hourly July estimates during years with observed decreases in July mean soil moisture below model wilting point marked as "SWC < $W_{wlt}$" compared to other years when July mean soil moisture was above model wilting point marked as "SWC > $W_{wlt}$". Bottom panel shows box plots of estimates during the winter, spring, and summer at Ramat Hanadiv. The boxes display the interquartile range (IQR) with the median marked with a horizontal line inside the box. The whiskers extend 1.5 IQR on either side of the box. Outliers were removed.





the relationship between net photosynthesis and stomatal conductance, and the conductance/resistance under ideal growing conditions. We discuss the sensitivity to model parameter values for all sites first and then focus on our case studies.

### 3.3 Moisture stress related parameter perturbations

The agreement between single-point modeled and flux-based $eg_s$ is more sensitive to the choice of values for certain parameters
compared to others, and certain parameters are only relevant at some sites while other parameter values can impact agreement at all sites (Figure 3 and Figure S2). At all sites, the perturbation of the $G_{s,H_2O,max}$ parameter used to set a maximum stomatal conductance under ideal growing conditions leads to the highest change in $MAD_{PM}$ and $MAD_{MED}$ with a change in parameter value (Figure 3 and Figure S2). The agreement between single-point modeled and flux-based $eg_s$ is sensitive to changes in $B_{VPD}$, $VPD_{max}$, $f(VPD)$, $f(RH_l)$, and $g_1$ at all sites investigated with sensitivity simulations (Figure 3 and
Figure S2). Conversely, perturbing the soil moisture stress parameters, $W_{wlt}$, $\psi_{soil,wlt}$, and $B$, result in substantially greater changes in $MAD_{PM}$ and $MAD_{MED}$ with changes in parameter values at sites that experienced substantial declines in soil moisture compared to sites that did not (Figure 3 and Figure S2). This indicates that the agreement between single-point modeled and flux-based $eg_s$ is sensitive to $VPD_{air}$ and $RH_l$ moisture stress parameters and functions at all sites while the sensitivity to soil moisture stress parameters is limited to sites that experience substantial soil moisture declines. Perturbing the
values of $R_{s,H_2O,ideal}$ and $\psi_{soil,wlt}$ results in the smallest changes in $MAD_{PM}$ and $MAD_{MED}$ (Figure 5 and Figure S2). While there were small changes in $MAD_{PM}$ and $MAD_{MED}$ with changes in $\psi_{soil,wlt}$ within the range of $\psi_{soil,wlt}$ values that was used to compute $\frac{\Delta MAD_{PM}}{\Delta v}$ and $\frac{\Delta MAD_{MED}}{\Delta v}$, perturbing the $\psi_{soil,wlt}$ to a substantially lower value, -1E09, results in a substantial reduction of bias during our case studies which we discuss below. Parameters perturbations that resulted in a near-zero $\frac{\Delta MAD_{PM}}{\Delta v}$ at all sites are not shown in Figure 3.
During July 2011 and 2012 when Borden Forest experienced reductions in soil moisture without experiencing reductions in flux-based $eg_s$, we find that decreasing the strength of the soil moisture stress by changing the values of the wilting point in both soil volumetric water content ($W_{wlt}$) and soil matric potential ($\psi_{soil,wlt}$) results in the largest reductions in absolute values of $MD_{eg_s,PM}$ and $MD_{eg_s,MED}$ (Figure 4 and Figure S3) for the models that showed considerable declines in $eg_s$ from the single-point model base simulations during July 2011 and 2012 (Figure 2). We discuss some notable examples of reductions
in absolute $MD_{eg_s,PM}$, but we keep the positive or negative sign to indicate the single-point model underestimation in the base simulations. Perturbing the wilting point resulted in a reduction in the 2011 and 2012 multiyear July $MD_{eg_s,PM}$ from 0.199 $cm\ s^{-1}$ from the base simulation for CMAQ STAGE to 0.021 $cm\ s^{-1}$ from the sensitivity simulation with the lowest $MD_{eg_s,PM}$ (Figure 4). Perturbing the wilting point resulted in a reduction in the 2011 and 2012 multiyear July $MD_{eg_s,PM}$ from 0.236 to 0.040 $cm\ s^{-1}$ for DO$_3$SE psn, a reduction from 0.270 to 0.148 $cm\ s^{-1}$ for DO$_3$SE multi, a reduction from 0.191
to -0.008 $cm\ s^{-1}$ for MLC-CHEM, a reduction from 0.295 to -0.083 $cm\ s^{-1}$ for TEMIR Wesely BB, and a reduction from 0.273 to -0.095 $cm\ s^{-1}$ for IFS SUMO Wesely (Figure 4).

For the second case study at Ramat Hanadiv, we find that $MAD_{PM}$ and $MAD_{MED}$ were the most sensitive during the dry summer months to the values of most of the parameters we studied (Figure 3 and Figure S2). Increasing the strength of the soil moisture stress through changes in the $\psi_{soil,wlt}$ and $B$ parameters and increasing the strength of the $VPD_{air}$ and $RH$



**Figure 3.** Comparisons of the change in median absolute difference between single point modeled $eg_s$ and flux-based $eg_{s,PM}$ ($\Delta MAD_{PM}$) with changes in a parameter or stress function value ($\Delta v$) for each parameter and stress function at each site. For each model-parameter pair or model-stress function pair, one summer $\frac{\Delta MAD_{PM}}{\Delta v}$ was calculated for Harvard Forest (HF), Borden Forest (BF), Ispra, (IS), and Hyytiälä (HY), and three $\frac{\Delta MAD_{PM}}{\Delta v}$ were calculated for Ramat Hanadiv: winter (RH-W), spring (RH-Sp), and summer (RH-S). $MAD_{PM}$ was calculated using daytime (half-) hourly estimates of $eg_s$.







**Figure 4.** The 2011 and 2012 multiyear monthly median difference between single-point modeled $eg_s$ and observed flux-based $eg_{s,PM}$ ($MD_{eg_{s,PM}}$) at Borden Forest for base and sensitivity simulations of single-point models. Sensitivity simulations perturbed the values of each parameter and stress function. Lines with filled dots show the $MD_{eg_{s,PM}}$ for base simulations of single-point models. Lines with open dots show the $MD_{eg_{s,PM}}$ for each parameter or stress function perturbation where each line represents one perturbation. Table 2 lists the interpretation of the parameters, stress functions, and the values used for sensitivity simulations. $W_{wlt}$ and $R_{s,H_2O,ideal}$ are shared among many models, and they are displayed in multiple plots to avoid plotting many model results in a single plot.





stress through changes in the $B_{VPD}$ parameter and the $f(VPD)$ and $f(RH_l)$ stress functions resulted in large reductions in absolute $MD_{eg_s,PM}$ and $MD_{eg_s,MED}$ during the dry summer months (Figure 5 and Figure S4). For most net-photosynthesis coupled models, changing the $g_1$ parameter also resulted in reductions in absolute $MD_{eg_s,PM}$ and $MD_{eg_s,MED}$ during the dry season (Figure 5 and Figure S4). We focus on July to discuss notable examples of the reductions in absolute $MD_{eg_s,PM}$ from the base simulations to the sensitivity simulations, but again, we keep the positive or negative sign to indicate the single-point

model overestimation in the base simulations.

Perturbing the $\psi_{soil,wlt}$ and $B$ parameters resulted in a $MD_{eg_s,PM}$ reduction from -0.228 to 0.030 $cm\ s^{-1}$ for TEMIR Wesely BB (Figure 5). Perturbing the $B_{VPD}$ parameter resulted in a $MD_{eg_s,PM}$ reduction from -0.276 to -0.054 $cm\ s^{-1}$ for TEMIR Zhang and a reduction from -0.202 to -0.067 $cm\ s^{-1}$ for GEM-MACH Zhang (Figure 5). For GEM-MACH Wesely, both the $f(VPD)$ stress function and the $R_{s,H_2O,ideal}$ parameter resulted in large reductions in $MD_{eg_s,PM}$ and $MD_{eg_s,MED}$

(Figure 5 and Figure S4). Perturbing the $f(VPD)$ stress function resulted in an $MD_{eg_s,PM}$ reduction from -0.305 to -0.037 $cm\ s^{-1}$, and perturbing the $R_{s,H_2O,ideal}$ parameter resulted in reduction in $MD_{eg_s,PM}$ from -0.305 to -0.043 $cm\ s^{-1}$ for GEM-MACH Wesely. Perturbing the $f(RH_l)$ stress function resulted in an $MD_{eg_s,PM}$ reduction from -0.152 to -0.003 $cm\ s^{-1}$ for CMAQ STAGE. Perturbing the $g_{1,BB}$ parameter resulted in a $MD_{eg_s,PM}$ reduction from -0.228 to -0.046 $cm\ s^{-1}$ for TEMIR Wesely BB (Figure 5). Perturbing the $g_{1,M}$ parameter resulted in a $MD_{eg_s,PM}$ reduction from -0.372 to -0.077 $cm\ s^{-1}$ for

TEMIR Wesely Medlyn (Figure 5).

## 4 Discussion

After comparing the single-point simulated stomatal component of $O_3$ $V_d$ from various dry deposition schemes implemented in 3D atmospheric chemistry models with inversion-based estimates using observed $\lambda E$ and $CO_2$ flux, we find that the agreement between single-point modeled and flux-based $eg_s$ is sensitive to the specification of moisture stress on stomatal conductance.

This is evident in both over- and underestimating the strength of stomatal moisture stress depending on ecosystem. We first discuss the two case studies: 1. Borden Forest, a temperate-boreal transition forest where observed declines in soil moisture did not limit the $GPP$, $\lambda E$, and stomatal conductance, and 2. Ramat Hanadiv, an eastern Mediterranean shrubland where the local vegetation is adapted to seasonal declines in soil moisture (Väänänen et al., 2020). Finally, we discuss the divergence between the flux-based estimates that we found at these two sites.

### 4.1 Moisture stress for stomatal conductance at a northern boreal-temperate transition forest

The case studies of observed declines in soil moisture at 50 cm depth at Borden Forest suggest that many single-point models struggle to capture the observed flux-based response of stomatal conductance during these conditions resulting in disagreements in $eg_s$. Long-term observations suggest that summertime net ecosystem productivity at Borden Forest is more strongly controlled by photosynthetically active radiation, air temperature, and soil temperature rather than soil moisture or $VPD_{air}$

(Froelich et al., 2015). High air and soil temperatures along with high photosynthetically active radiation during the summer months coincide with the highest net ecosystem productivity (Froelich et al., 2015). July 2011 and 2012 were months with






**Figure 5.** Monthly median difference between single-point modeled $eg_s$ and observed flux-based $eg_{s,PM}$ ($MD_{eg_{s,PM}}$) at Ramat Hanadiv for base and sensitivity simulations of single-point models. Some months have multiple years of data. Sensitivity simulations perturbed the values of each parameter and stress function. Lines with filled dots show the $MD_{eg_{s,PM}}$ for base simulations of single-point models. Lines with open dots show the $MD_{eg_{s,PM}}$ for each parameter or stress function perturbation where each line represents one perturbation. Table 2 lists the interpretation of the parameters, stress functions, and the values used for sensitivity simulations. $W_{wlt}$ and $R_{s,H_2O,ideal}$ are shared among many models, and they are displayed in multiple plots to avoid plotting many model results in a single plot.




observed declines in soil moisture at 50 cm depth in this study, but interannualy, July 2011 and 2012 were also the year with the highest shortwave radiation and $GPP$ at Borden Forest during the summer months (Figure S1).

This suggests that the observed decline in soil moisture at 50 cm depth does not limit stomatal conductance. Previous
analysis of long-term $CO_2$ exchange data from Borden Forest from 1996 - 2013 showed that the only year when the drops in soil moisture and precipitation were severe enough to create noticeable declines in $GPP$ was 2007 (Froelich et al., 2015). This indicates that while the wilting point specified for 50 cm depth in some single-point models estimated the lowest $eg_s$ during July 2011 and 2012 at Borden Forest, the observed flux-based estimates of stomatal conductance and previous longer term estimates of $GPP$ do not consider 2011 and 2012 to be years of vegetation water stress in terms of reductions in $GPP$. A wilting point
specified for measurements of soil moisture at 50 cm depth depth might not reflect the soil water sources that are available to the trees at Borden Forest where red maple (*Acer rubrum*) make up to 50% of the tree species composition (Teklemariam et al., 2009). At the Hubbard Brook Experimental Forest (HBEF) in northeastern United States, red maples primarily used shallow soil water sources at less than 10 cm depth during June 2018 and less than 30 cm depth during July 2018 (Harrison et al., 2020). However, their primary source of soil water shifted to depths of 90 - 100 cm in August suggesting that red maples can
switch the depths from which they access soil water within a growing season (Harrison et al., 2020). Seasonal adjustments in tree water uptake depths have been widely observed (Bachofen et al., 2024), and the possibility of similar dynamics in tree access to soil water at Borden Forest makes it challenging to apply wilting point type thresholds at a single measurement depth for point models. Disagreements between single-point modeled and observed flux-based $eg_s$ suggest that simulating the effects of soil moisture on stomatal conductance for a northern boreal-temperate transition forest can benefit from incorporating tree
access to variable soil water sources.

## 4.2    Moisture stress for stomatal conductance at a seasonally dry eastern Mediterranean shrubland

There can be significant inter-specific variation in net photosynthesis, transpiration, and stomatal conductance among the co-existing woody species at Ramat Hanadiv during the dry summer months due to varying drought resistance between the species (Väänänen et al., 2020). Native woody species at this shrubland like *Quercus calliprinos* employ a host of resistance
strategies and likely exhibit greater rooting depth and access to deep water reserves to withstand seasonal declines in soil moisture (Väänänen et al., 2020). The leaf-level stomatal conductance of the woody species declines during the dry summer months with increases in $VPD_{air}$ and decreases in soil moisture (Väänänen et al., 2020). Our flux-based estimates of $eg_s$ and previous flux-based estimates in the region also confirm that the vegetation at Ramat Hanadiv and other regional sites experience declines in stomatal conductance into the dry summer months as soil moisture declines and $VPD_{air}$ increases (Li
et al., 2019, 2018).

Many single-point models simulated increasing $eg_s$ into the dry months. These models include those that simulate the effects of both soil moisture and $VPD_{air}$ or $RH_l$, those that simulate the effects of $VPD_{air}$ and leaf water potential, and those that do not simulate the effects of moisture stress on stomatal conductance. Increasing the $RH_l$ and $VPD$ stress through parameter and stress function perturbations increased the summertime agreement between single-point modeled and flux-based $eg_s$ for
models that simulated increasing $eg_s$ into summer. Some single-point models that simulate increasing $eg_s$ into the dry months



prescribe a seasonally varying $R_{s,H_2O,ideal}$. The high wintertime and low summertime $R_{s,H_2O,ideal}$ in these models, based on temperate ecosystems, likely contributes to disagreements in the seasonality of $eg_s$ with other models and flux-based estimates. Finally, the models that simulate the effects of leaf water potential on stomatal conductance without simulating the effects of soil water potential also simulated increasing $eg_s$ into the dry summer months. Leaf water potential is simulated to vary only

as a function of shortwave radiation (Clifton et al., 2023). However, leaf water potential is mechanistically coupled with soil water potential although the relationship between the two can vary due to plant water regulation (Sack and Holbrook, 2006; Martínez-Vilalta et al., 2014; Venturas et al., 2017). The leaf water potential of woody species at Ramat Hanadiv has shown strong linear relationships with soil water potential (Väänänen et al., 2020). The misrepresentation of leaf water potential likely contributed to the disagreement between single-point modeled and flux-based $eg_s$ in the dry months. Coupling the simulation

of leaf water potential with the simulation of soil water potential in these models presents an opportunity to improve stomatal conductance and by extension $eg_s$ sensitivity to drought.

### 4.3 Comparison of available methods to estimate the stomatal component of $O_3$ dry deposition from observed latent heat and $CO_2$ flux

We found that individual $eg_s$ estimates calculated from inversion and observed flux-based methods can exhibit disagreements

in two ecosystems. For example, $eg_s$ estimates from a Penman-Montieth inversion using latent heat flux can disagree in $eg_s$ from fitting an optimality-based stomatal conductance model using $GPP$ partitioned from $NEE$ at Borden Forest. We also found disagreements in magnitude between the two methods to infer stomatal conductance at Ramat Hanadiv. The source of this disagreement is the difference in the underlying flux used to calculate the stomatal conductance estimate and the dependence of conductance on $VPD$ in the method used. At all sites, $G_{s,H_2O,MED}$ had a higher dependence on $GPP$ and $NEE$ compared

to the dependence of $G_{s,H_2O,PM}$ estimates on latent heat flux used in the Penman-Montieth inversion.

The dependence of stomatal conductance on $VPD_{canopy-air}$ indicated that at low $VPD_{canopy-air}$, higher stomatal conductance was estimated from a PM inversion compared to upscaling the Medlyn et al. (2011) model using $GPP$. This could explain the disagreements at Borden Forest between the two flux-based estimates. It is important to note that we used the nighttime method to partition $NEE$ into $GPP$ and $R_{eco}$ to avoid the added dependence of $GPP$ on near-surface $VPD$ that

the daytime method introduces (Lasslop et al., 2010) considering the Medlyn et al. (2011) model also includes the effects of $VPD_{canopy-air}$. Regardless of the varying degrees of dependence on $VPD$ that single-point models and flux-based estimates can display, the two major findings from the two case studies at Borden Forest and Ramat Hanadiv hold. At Ramat Hanadiv, using both $GPP$ and latent heat flux to estimate stomatal conductance shows a decline of $eg_s$ into the dry summer months which is confirmed by previous leaf-level gas exchange measurements at the site. Furthermore, both flux-based estimates of

$eg_s$ do not suggest a substantial decline in $eg_s$ simulated by single-point models during July 2011 and 2012 at Borden Forest.



## 5 Conclusions

Simulating the dry deposition of $O_3$ to the land surface is a crucial component of simulating $O_3$ concentrations and air quality. Here, we compared estimates from various dry deposition schemes implemented in chemical transport models run as single-point models forced with observed micro-meteorology and environmental conditions at flux tower sites. Specifically, we focused on comparing observed flux-based estimates of the stomatal component of $O_3$ dry deposition, $eg_s$, with simulations of $eg_s$ by the single-point models by aiming to answer three research questions:

1. How does the stomatal component of $O_3$ deposition velocity from single-point models compare with the estimates from observed latent heat and $CO_2$ flux?

2. How does the specification of moisture stress in single-point models impact the agreement in the stomatal component of $O_3$ deposition velocity between single-point models simulations and flux-based estimates during times of increased atmospheric or soil dryness?

3. In dry deposition schemes, how do stomatal moisture stress parameters affect the agreement between single-point models and observed flux-based estimates?

To answer question 1, we find that monthly mean observed flux-based $eg_s$ agree with a central ensemble range of monthly mean $eg_s$ by single-point model simulations during parts of the growing season at all sites when multiyear data was available. However, when we focused on specific cases of increased atmospheric or soil dryness within the observational dataset, we found that moisture stress specification resulted in disagreements between single-point modeled and flux-based $eg_s$. To answer question 2, we find that single-point modeled soil moisture stress for stomatal conductance was too strong in a light and temperature limited northern temperate-boreal transition forest where high summertime photosynthetically active radiation and temperatures favor high net ecosystem productivity and the tree species likely have access to deeper soil water sources compared to the depth at which the soil moisture was measured. This resulted in underestimation of $eg_s$ by some single-point models compared to observed flux-based estimates because observed soil moisture at 50 cm depth fell below the model specified wilting point. Furthermore, an eastern Mediterranean shrubland where seasonality in stomatal conductance is driven by water availability is poorly represented by some single-point models. Many single-point models overestimated $eg_s$ compared to observed flux-based estimates during the dry summer months.

To answer question 3, we find that at all sites examined, single-point modeled $eg_s$ and the agreement with observed flux-based $eg_s$ was sensitive to parameters that control the vapor pressure deficit stress and the relationship between net photosynthesis and stomatal conductance. Conversely, the simulation of $eg_s$ and the agreement with observed flux-based $eg_s$ was highly sensitive to parameters that control the soil moisture stress only at specific sites that experienced substantial declines in soil moisture. This suggests that the simulated $eg_s$ is highly sensitive to parameter choice for soil moisture stress when environmental conditions start to reach model thresholds like wilting point leading to large disagreements between single point modeled and flux-based $eg_s$. Clifton et al. (2023) showed that stomatal conductance is the most important driver of the the seasonal variability in simulated deposition velocity in many of the single-point models at all of the sites in this study. Thus, the simulations of stomatal conductance will directly impact total deposition velocity among these models. This indicates that the




impacts of simulating stomatal moisture stress on stomatal conductance and $eg_s$ estimates shown here could likely propagate
to O$_3$ deposition velocity.

In order to simulate O$_3$ dry deposition in the face of projected increases in aridity, understanding and correctly parameterizing
the response of stomatal conductance to decreases in soil moisture and increased vapor pressure deficit across a range of
ecosystems will play a key role in improving the simulation of O$_3$ dry deposition during drought conditions. Other beneficial

model developments could include simulating the impact of O$_3$ on stomatal conductance. O$_3$ uptake itself can impact stomatal
conductance and its response to other environmental conditions like soil moisture. Using results from chamber and free air
controlled exposure studies, various methods to incorporate O$_3$ effects on stomata in the Ball-Woodrow-Berry (1987) model,
the Medlyn et al. (2011) model, and Jarvis et al. (1976) type models have been introduced (Lombardozzi et al., 2012a, 2015;
Hoshika et al., 2020, 2015, 2018), but there is yet to be an analysis of how such model additions would impact O$_3$ dry

deposition. Ongoing developments in land surface modeling of stomatal conductance and vegetation responses to water stress
will likely benefit components of tropospheric O$_3$ modeling.

*Data availability.* The observed flux and meteorological forcing datasets are available to individuals wishing to participate in this effort on
a password-protected site managed by the United States Environmental Protection Agency, subject to the individual's agreement that the
people who created and maintained the observation datasets are included in publications as the people see fit. The flux data from Harvard

Forest is publicly available at https://doi.org/10.6073/pasta/56c6fe02a07e8a8aaff44a43a9d9a6a5 and
https://harvardforest1.fas.harvard.edu/exist/apps/datasets/showData.html?id=HF004. The flux data from Borden Forest is publicly available
at https://ameriflux.lbl.gov/sites/siteinfo/CA-Cbo. The flux data from Hyytiälä is publicly available at https://smear.avaa.csc.fi/download.
Flux tower data from Ramat Hanadiv will be available to the European Fluxes Database at http://www.europe-fluxdata.eu/.

*Author contributions.* AMK developed the manuscript's research questions and methods with feedback from OEC and PCS, wrote and re-

vised the manuscript, performed calculations for flux-based $eg_s$, and conducted the analysis. CH assisted with data processing, model base
and sensitivity simulations, and coordination among authors. PCS, OEC, CH, SG, LG, ET, TW, IM, SJS, LH, and SS provided manuscript
revisions. JOB contributed CMAQ STAGE base and sensitivity simulations. LE, NB, and SB contributed DO$_3$SE base and sensitivity sim-
ulations. PC and PAM contributed GEM-MACH base and sensitivity simulations. JF contributed IFS base and sensitivity simulations. EF,
QL, and ET contributed data from Ramat Hanadiv. LG contributed MLC-CHEM base and sensitivity simulations and provided feedback

throughout manuscript development. OG, IG, and GM contributed data from Ispra. CDH provided GEOS-Chem base and sensitivity simu-
lations. LH and TW contributed data from Bugacpuszta. VH contributed IFS base and sensitivity simulations. IM and TV contributed data
from Hyytiälä. JWM contributed data from Harvard Forest. JLPC and RSJ contributed WRF-Chem base and sensitivity simulations. JP and
LR contributed M3Dry base simulations. RS, ZW, and LZ contributed data from Borden Forest. SS and APKT contributed TEMIR base and
sensitivity simulations. All authors contributed to manuscript writing and useful discussions on data analysis and processing and results.



*Competing interests.* At least one of the (co-)authors is a member of the editorial board of Atmospheric Chemistry and Physics. The authors have no other competing interests to declare.

*Disclaimer.* The views expressed in this article are those of the authors and do not necessarily represent the views or policies of the U.S. Environmental Protection Agency.

*Acknowledgements.* AMK and PCS acknowledge support from the U.S. National Science Foundation Macrosystems Biology award 2106012

and the University of Wisconsin-Madison Office of Vice Chancellor for Research and Graduate Education with funding from the Wisconsin Alumni Research Foundation. LH acknowledges support from the Sustainable Development and Technologies National Programme of the Hungarian Academy of Sciences (FFT NP FTA) and by the Hungarian Research and Technology Innovation Fund (OTKA) project no. K-138176. APKT acknowledges support from the Collaborative Research Fund (Ref. No.: C5062-21GF) and from the Research Grants Council of Hong Kong. ET acknowledges support from the Israel Science Foundation, Grant No. 543/22. JWM acknowledges support for

the Harvard Forest flux tower. The Harvard Forest flux tower is a component of the Harvard Forest LTER site, supported by the National Science Foundation, and an AmeriFlux core site supported by the AmeriFlux Management Project with funding from the U.S Department of Energy.



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
