# Peer review of "Ozone dry deposition through plant stomata: Multi-model comparison with flux observations and the role of water stress as part of AQMEII4 Activity 2"

_EGUsphere, 2024_

## Referee Comment (RC2)

**Ozone dry deposition through plant stomata: Multi-model comparison with flux observations and the role of water stress as part of AQMEII4 Activity 2**

Khan et al., EGUsphere [preprint], https://doi.org/10.5194/egusphere-2024-3038

**General comments**

Dry deposition is an important removal process of trace gases and aerosols from the atmosphere to the Earth's surface. This paper reports on an intercomparison study of 18 dry deposition schemes used in current air pollution and atmospheric chemistry transport models, as part of the Air Quality Model Evaluation International Initiative 4 (AQMEII4), Activity 2. The paper is part of a Special Issue on AQMEII4.

This paper by Khan et al. follows on from a preceding paper in the same Special Issue (cited paper by Clifton et al., 2023), which evaluates the overall dry deposition process for ozone ($O_3$), through comparisons of modelled and observed deposition velocities. Clifton et al. found "models can disagree with respect to relative contributions from the [stomatal and non-stomatal] pathways, even when they predict similar deposition velocities, or agree with respect to the relative contributions but predict different deposition velocities". This paper extends the analysis, considering the stomatal component and investigates through two case studies how the stomatal uptake of ozone responds to moisture stress.

This is a very detailed paper, with a focus again on ozone ($O_3$). There are measured $O_3$ deposition fluxes, albeit from a limited number of sites (Figure 2 in cited paper by Clifton et al., 2020). The analysis is based on six of these sites in the Northern Hemisphere: boreal, temperate, and temperate-boreal transition forests (4 sites), together with an eastern Mediterranean shrubland (1) and a temperate grassland (1) site.

The authors aim to address the performance of the different schemes, process representation and sensitivity to parameter values. The paper reads wells and is likely to be of wider interest. The process-based approaches for stomatal conductance are used in land surface models (which form the land surface component of climate and Earth System models). I recommend publication after addressing the following comments.

**Specific Comments**

1. Dry deposition schemes

    The authors group the deposition schemes into 4 main types:
    - net photosynthesis coupled models (NP:SM/VPD/RH)
    - Jarvis-type models that include both soil and air moisture impacts (J:SM/VPD/RH)
    - Jarvis-type models that only include $VPD_{air}$ (J:VPD)
    - Jarvis-type models that do not include soil moisture or air moisture impacts (J:NoSM/VPD/RH)

    Details of the dry deposition schemes are provided in the preceding paper by Clifton et al. (2023). While Table 2 and Figure 1 do have relevant information, it would be helpful to have a short summary table in this paper, to list the dry deposition schemes and the scheme 'type' to which it belongs.

2.  Comparison to observations and/or observation-derived parameters

The approach taken by Clifton et al. (2023) is also used here, i.e. the dry deposition schemes are run as point versions using measured meteorological and environmental variables from the 6 sites.

The authors use two approaches to derive estimates of $O_3$ stomatal conductance from the observed fluxes of latent heat and the net ecosystem exchange of $CO_2$: inversion of the evaporation-resistance form of the Penman-Monteith (PM) equation and fitting the stomatal conductance model of Medlyn et al. So far, this is 'standard' analysis of flux data. Is there a reason why the $O_3$ flux observations could not be used to give the stomatal $O_3$ component? It is usually assumed that the night-time measurements give the non-stomatal component.

I could understand the approach used here if the intention is an evaluation of stomatal conductance schemes in general and not of $O_3$ stomatal conductance specifically. Further, with the approach adopted, more FluxNet sites could be used and for a wider range of site/vegetation types. Some comment or justification is needed.

From Figure 1, the model ensemble central range seems to reproduce observations at the forested sites, although the peak conductance appears to occur later in the year at Hyytiälä (Figure 1). There are greater differences at the grassland and shrubland sites. There seems to be some evidence that the net-photosynthesis type models perform better. Is this the case?

3.  Case studies

Two case studies are investigated to understand the impact of moisture stress on stomatal conductance, using observations from the Borden Forest and Ramat Hanadiv sites. Sensitivity studies are then undertaken, varying the values of parameters that control moisture stress. The analysis indicates the need to include more processes in the deposition schemes, e.g. inclusion of rooting depth. These are a valuable part of the study.

4.  Conclusions and wider interest

While areas for future development are Identified (inclusion of rooting depth), there are no recommendations about the relative merits or performance of the different types of deposition schemes (i.e. NP:SM/VPD/RH, J:SM/VPD/RH, J: VPD and J:NoSM/VPD/RH). Arguably, some of the Jarvis type schemes do not include all the factors that control stomatal exchange, but this may well be compensated by calibration and choice of parameter values. Can the authors say or give some indications if one type is preferable?

$O_3$ vegetation damage is mentioned. This needs some clarification. Presumably, the authors are implying that parameter values may need adjusting to account for $O_3$ damage. I am aware that at least one leading land surface model (the UK model JULES) includes $O_3$ vegetation damage (Sitch et al., 2007; Oliver et al., 2018).

The feedback between increased $CO_2$ concentrations leading to changing plant physiology and climate has long been known (e.g. Seneviratne et al., 2010; Betts et al., 2007). Further, vegetation water and drought stress are of current interest to the land surface modelling community (Williams et al., 2019, Harper et al., 2021). Therefore, I agree that "ongoing developments in land surface modelling of stomatal conductance and vegetation responses to water stress will likely benefit components of tropospheric $O_3$ modelling". There needs to be more engagement between the air pollution and land surface modelling community.

**Technical Corrections**

- Line 201: missing definite article in "from observed CO2 flux" -> "from the observed CO2 flux".
- Line 465: delete duplicate "depth" in "soil moisture at 50 cm depth  might"
- Lines 552-553: delete duplicate "the" in "most important driver of the  seasonal variability

**Data availability**

Information is provided.

**Additional references:**

Betts et al., 2007: Projected increase in continental runoff due to plant responses to increasing carbon dioxide. Nature 448, 1037-1041, https://doi.org/10.1038/nature06045.

Harper et al., 2021: Improvement of modeling plant responses to low soil moisture in JULESvn4.9 and evaluation against flux tower measurements, Geosci. Model Dev., 14, 3269-3294, https://doi.org/10.5194/gmd-14-3269-2021.

Oliver et al., 2018: Large but decreasing effect of ozone on the European carbon sink, Biogeosciences, 15, 4245–4269, https://doi.org/10.5194/bg-15-4245-2018.

Seneviratne et al., 2010: Investigating soil moisture-climate interactions in a changing climate: A review. Earth-Science Reviews, 99, 125-161. https://www.sciencedirect.com/science/article/pii/S0012825210000139.

Sitch et al., 2007: Indirect radiative forcing of climate change through ozone effects on the land-carbon sink. Nature 448, 791-794, https://doi.org/10.1038/nature06059.

Williams et al., 2019: How can the First ISLSCP Field Experiment contribute to present-day efforts to evaluate water stress in JULESv5.0?, Geosci. Model Dev., 12, 3207–3240, https://doi.org/10.5194/gmd-12-3207-2019.

---

## Author Response (AR1)

**Response to Referee 1 comments for the submitted manuscript:  Ozone dry deposition through plant stomata: Multi-model comparison with flux observations and the role of water stress as part of AQMEII4 Activity 2**

Khan et al.

We thank the referee for their insightful and helpful comments. We have made several changes in response to the comments provided. Below, we respond to each of the referee's comments.

Referee comments and author response.

This manuscript presents a multi-model comparison of $O_3$ dry deposition from chemistry climate models with the same observed flux-based estimates at six locations in the Northern Hemisphere. The observational dataset includes a good number of station sites distributed across continents with a good temporal resolution and are or can become available under request. The methodology followed is well explained and properly referenced. Results are clearly presented and discussed. Comparisons with other results in existing relevant literature and the characteristics of each evaluated site justify their findings. The authors made a comprehensive revision of previous studies including stomatal conductance models. The manuscript tries to fill a gap in our knowledge and is relevant for the scientific community, especially for atmospheric chemistry modelers. I only have some suggestions that will improve the quality and readability of the manuscript.

1. Line 147, equation (5): the authors explained all the terms in the equation except for B. Could the authors briefly describe what B represents?

   **Response:** We have added an explanation of the B parameter. This can be found at line 145 of the revised manuscript.

2. In lines 259 and 260, it is said that the number of sensitivity simulations depends on the model, which can be seen in Figures 4 and 5, and after that, the authors explain the reason behind that. However, I think the manuscript

would be clearer if the number of sensitivity tests and the perturbed parameters per model were summarized in a Table.

**Response:** We have added this information to tables in the supplement.

3. On a related note, Table 2 lists the parameters perturbed in the sensitivity test with their corresponding range of values. However, the reader does not know the default values for each parameter and model nor the magnitude of the perturbations with respect to that default value. I think it would be interesting to see this information, perhaps adding it to the table I suggested in point 2.

   **Response:** We have added this information to tables in the supplement.

4. Line 399: it is the first time that $B_{VPD}$ appears in the text, but it is not explained what it means. Please, include a brief description.

   **Response:** We have added an explanation of the $B_{VPD}$ parameter at line 401 of the revised manuscript.

5. Line 405: wrong reference to Figure 5 (it should be Figure 3).

   **Response:** We have corrected the figure reference. The correction can be found at line 407 of the revised manuscript.

6. Lines 419 and 420: "…a reduction from 0.191 to -0.008 cm s−1 for MLC-CHEM…". Please, revise either the values in the text or the values plotted in Figure 4. In that figure, the values of the wilting point for MLC-CHEM in July go from around 0.2 to something close to -0.2, if I understood correctly.

   **Response:** By "reduction in median difference" we mean the reduction to a median difference closest to 0 from the median difference of the base simulations. The correct way to interpret Figures 4-5 is to look at the reduction in the median difference from the bold lines with filled circles to the dashed line with open circles that is closest to 0.

7. Line 465: "..50 cm depth depth..", please, delete the repeated "depth".

   **Response:** We have made the correction at line 481 of the revised manuscript.

8. Line 552: "...the most important driver of the the...", please, delete the repeated "the".

   **Response:** We have made the correction at line 567 of the revised manuscript.

**Response to Referee 2 comments for the submitted manuscript:  Ozone dry deposition through plant stomata: Multi-model comparison with flux observations and the role of water stress as part of AQMEII4 Activity 2**

Khan et al.

We thank the referee for their insightful and helpful comments. We have made changes in response to the comments provided. Below, we respond to each comment from referee 2.

Referee comments and author response

Dry deposition is an important removal process of trace gases and aerosols from the atmosphere to the Earth's surface. This paper reports on an intercomparison study of 18 dry deposition schemes used in current air pollution and atmospheric chemistry transport models, as part of the Air Quality Model Evaluation International Initiative 4 (AQMEII4), Activity 2. The paper is part of a Special Issue on AQMEII4. This paper by Khan et al. follows on from a preceding paper in the same Special Issue (cited paper by Clifton et al., 2023), which evaluates the overall dry deposition process for ozone ($O_3$), through comparisons of modelled and observed deposition velocities. Clifton et al. found "models can disagree with respect to relative contributions from the [stomatal and non-stomatal] pathways, even when they predict similar deposition velocities, or agree with respect to the relative contributions but predict different deposition velocities". This paper extends the analysis, considering the stomatal component and investigates through two case studies how the stomatal uptake of ozone responds to moisture stress.
This is a very detailed paper, with a focus again on ozone ($O_3$). There are measured $O_3$ deposition fluxes, albeit from a limited number of sites (Figure 2 in cited paper by Clifton et al., 2020). The analysis is based on six of these sites in the Northern Hemisphere: boreal, temperate, and temperate-boreal transition forests (4 sites), together with an eastern Mediterranean shrubland (1) and temperate grassland (1) site. The authors aim to address the performance of the different schemes, process representation and sensitivity to parameter values. The paper reads wells and is likely to be of wider interest. The process-based approaches for

stomatal conductance are used in land surface models (which form the land surface component of climate and Earth System models). I recommend publication after addressing the following comments.

1. Dry deposition schemes

    The authors group the deposition schemes into 4 main types:

    • net photosynthesis coupled models (NP:SM/VPD/RH)

    • Jarvis-type models that include both soil and air moisture impacts (J:SM/VPD/RH)

    • Jarvis-type models that only include VPDair (J:VPD)

    • Jarvis-type models that do not include soil moisture or air moisture impacts (J:NoSM/VPD/RH)

    Details of the dry deposition schemes are provided in the preceding paper by Clifton et al. (2023). While Table 2 and Figure 1 do have relevant information, it would be helpful to have a short summary table in this paper, to list the dry deposition schemes and the scheme 'type' to which it belongs.

    **Response:** As a revision to make things more helpful, we have added another column to table 2 to show the exact $G_s$ scheme type a single point model belongs to in addition to that information being displayed in Figure 1.

2. The approach taken by Clifton et al. (2023) is also used here, i.e. the dry deposition schemes are run as point versions using measured meteorological and environmental variables from the 6 sites. The authors use two approaches to derive estimates of $O_3$ stomatal conductance from the observed fluxes of latent heat and the net ecosystem exchange of $CO_2$: inversion of the evaporation-resistance form of the Penman-Monteith (PM) equation and fitting the stomatal conductance model of Medlyn et al. So far, this is 'standard' analysis of flux data. Is there a reason why the $O_3$ flux observations could not be used to give the stomatal $O_3$ component? It is usually assumed that the night-time measurements give the non-stomatal component. I could understand the approach used here if the intention is an evaluation of stomatal conductance schemes in general and not of O3 stomatal conductance specifically. Further, with the approach adopted, more FluxNet sites could be used and for a wider range of site/vegetation types.

Some comment or justification is needed. From Figure 1, the model ensemble central range seems to reproduce observations at the forested sites, although the peak conductance appears to occur later in the year at Hyytiälä (Figure 1). There are greater differences at the grassland and shrubland sites. There seems to be some evidence that the net-photosynthesis type models perform better. Is this the case?

**Response:** The ozone ($O_3$) flux observations are used to give the stomatal component ($eg_s$) of $O_3$ deposition velocity, $V_d$. Equations 13 - 14 in the submitted manuscript described how $eg_s$ is calculated as a function of the $G_s$, canopy conductance ($G_c$), and $V_d$ of $O_3$. The $O_3$ flux observations are used to calculate the $O_3$ $V_d$. These three quantities require the use of latent heat, $CO_2$, and $O_3$ flux. As a revision, we have added the equation for $V_d$ to further clarify how the $O_3$ flux observations were used in the submitted manuscript. This added equation is listed as equation 13, and it can be found between lines 204 – 208 in the revised manuscript. Please note that the equations describing the calculation of $eg_s$ are listed as equations 14-15 in the revised manuscript.

The approach here specifically compares $eg_s$. The $eg_s$ quantity is different from $G_s$ even though it is calculated as a function of $G_s$. The approach of comparing $eg_s$ cannot be replicated at other FLUXNET sites because they lack $O_3$ flux observations, and thus $V_d$, $G_c$, and the resulting $eg_s$ cannot be calculated. Since this study builds on Clifton et al. (2023) which demonstrated that the relative contribution of each simulated depositional pathway to simulated $V_d$ can vary among models, we are concerned with $eg_s$ and not $G_s$ alone. An analysis of flux-based $G_s$ can be carried out across many FLUXNET sites. Our specific analysis is concerned with the relative contribution of $G_s$ to $V_d$ at the sites examined in Clifton et al. (2023), and it will lose much context if we do not know what observed $V_d$ is. We address the reviewer's final point from their second specific comment in our answer to their fourth specific comment.

3. Case studies

Two case studies are investigated to understand the impact of moisture stress on stomatal conductance, using observations from the Borden Forest and Ramat Hanadiv sites. Sensitivity studies are then undertaken, varying the values of parameters that control moisture stress. The analysis indicates the need to include more processes in the deposition schemes, e.g. inclusion of rooting depth. These are a valuable part of the study.

**Response:** We agree that targeted investigations about the impact of simulating vegetation moisture stress are important for improving simulations.

4. Conclusions and wider interest
   While areas for future development are Identified (inclusion of rooting depth), there are no recommendations about the relative merits or performance of the different types of deposition schemes (i.e. NP:SM/VPD/RH, J:SM/VPD/RH, J: VPD and J:NoSM/VPD/RH). Arguably, some of the Jarvis type schemes do not include all the factors that control stomatal exchange, but this may well be compensated by calibration and choice of parameter values. Can the authors say or give some indications if one type is preferable? $O_3$ vegetation damage is mentioned. This needs some clarification. Presumably, the authors are implying that parameter values may need adjusting to account for $O_3$ damage. I am aware that at least one leading land surface model (the UK model JULES) includes $O_3$ vegetation damage (Sitch et al., 2007; Oliver et al., 2018). The feedback between increased $CO_2$ concentrations leading to changing plant physiology and climate has long been known (e.g. Seneviratne et al., 2010; Betts et al., 2007). Further, vegetation water and drought stress are of current interest to the land surface modelling community (Williams et al., 2019, Harper et al., 2021). Therefore, I agree that "ongoing developments in land surface modelling of stomatal conductance and vegetation responses to water stress will likely benefit components of tropospheric $O_3$ modelling". There needs to be more engagement between the air pollution and land surface modelling community.

**Response:** We have added a discussion regarding the nuance behind choosing a preferential stomatal conductance model type. This discussion can be found at lines 446 – 461of the revised manuscript. There is no conclusive evidence that all models within one class of models are always better at all ecosystems in the study. However, the study does suggest that increased and improved process representation of moisture stress is beneficial for agreement across different ecosystems.

We have also included some more clarification of existing methods to incorporate $O_3$ impacts on stomatal conductance at lines 576 – 579 of the revised manuscript.

• Line 201: missing definite article in "from observed $CO_2$ flux" -> "from the observed $CO_2$ flux"

**Response:** We have made the correction, and it can be found at line 198 of the revised manuscript.

• Line 465: delete duplicate "depth" in "soil moisture at 50 cm depth depth might"

**Response:** We have made the correction, and it can be found at line 481 of the revised manuscript.

• Lines 552-553: delete duplicate "the" in "most important driver of the the seasonal variability

**Response:** We have made the correction, and it can be found at line 567 of the revised manuscript.

**References**

Betts et al., 2007: Projected increase in continental runoff due to plant responses to increasing carbon dioxide. Nature 448, 1037-1041, https://doi.org/10.1038/nature06045.

Clifton, O.E., Schwede, D., Hogrefe, C., Bash, J.O., Bland, S., Cheung, P., Coyle, M., Emberson, L., Flemming, J., Fredj, E., Galmarini, S., Ganzeveld, L., Gazetas, O., Goded, I., Holmes, C.D., Horváth, L., Huijnen, V., Li, Q., Makar, P.A., Mammarella, I., Manca, G., Munger, J.W., Pérez-Camanyo, J.L., Pleim, J., Ran, L., San Jose, R., Silva, S.J., Staebler, R., Sun, S., Tai, A.P.K., Tas, E., Vesala, T., Weidinger, T., Wu, Z., Zhang, L., 2023. A single-point modeling approach for the intercomparison and evaluation of ozone dry deposition across chemical transport models (Activity 2 of AQMEII4). Atmospheric Chemistry and Physics 23, 9911–9961. https://doi.org/10.5194/acp-23-9911-2023

Harper et al., 2021: Improvement of modeling plant responses to low soil moisture in JULESvn4.9 and evaluation against flux tower measurements, Geosci. Model Dev., 14, 3269-3294, https://doi.org/10.5194/gmd-14-3269-2021.

Oliver et al., 2018: Large but decreasing effect of ozone on the European carbon sink, Biogeosciences, 15, 4245–4269, https://doi.org/10.5194/bg-15-4245-2018.

Seneviratne et al., 2010: Investigating soil moisture-climate interactions in a changing climate: A review. Earth-Science Reviews, 99, 125-161. https://www.sciencedirect.com/science/article/pii/S0012825210000139.

Sitch et al., 2007: Indirect radiative forcing of climate change through ozone effects on the land-carbon sink. Nature 448, 791-794, https://doi.org/10.1038/nature06059.

Williams et al., 2019: How can the First ISLSCP Field Experiment contribute to present-day efforts to evaluate water stress in JULESv5.0?, Geosci. Model Dev., 12, 3207–3240, https://doi.org/10.5194/gmd-12-3207-2019.

---

## Referee Report (RR1)

**Ozone dry deposition through plant stomata: Multi-model comparison with flux observations and the role of water stress as part of AQMEII4 Activity 2**

Khan et al., EGUsphere [preprint], https://doi.org/10.5194/egusphere-2024-3038

**General comments**

I reviewed the original manuscript (Reviewer 2). The author response and the changes made to the manuscript have addressed the points raised in my review. I therefore recommend that the revised manuscript can be accepted as is.

**Specific Comments**

1. Dry deposition schemes: Additional information has been provided on the deposition schemes.

2. Comparison to observations and/or observation-derived parameters: The authors have clarified that ozone flux measurements are needed and this limits the number of sites that can be used.

4. Performance of deposition schemes and $O_3$ vegetation damage: While the authors do not make any specific recommendations about the relative merits or performance of the different types of deposition schemes (i.e. NP:SM/VPD/RH, J:SM/VPD/RH, J: VPD and J:NoSM/VPD/RH), they have added text on this point. The authors identify the need to account for soil moisture stress, at least, to improve the performance of these ozone deposition schemes.

   The authors have amended the text on $O_3$ vegetation damage and added relevant references.